# Causal Discovery from Conditionally Stationary Time Series

## Abstract

Causal discovery, i.e., inferring underlying causal relationships from observational data, has been shown to be highly challenging for AI systems. In time series modeling context, traditional causal discovery methods mainly consider constrained scenarios with fully observed variables and/or data from stationary time-series. We develop a causal discovery approach to handle a wide class of non-stationary time-series that are *conditionally stationary*, where the non-stationary behaviour is modeled as stationarity conditioned on a set of (possibly hidden) state variables. Named State-Dependent Causal Inference (SDCI), our approach is able to recover the underlying causal dependencies, provably with fully-observed states and empirically with hidden states. The latter is confirmed by experiments on synthetic linear system and nonlinear particle interaction data, where SDCI achieves superior performance over baseline causal discovery methods. Improved results over non-causal RNNs on modeling NBA player movements demonstrate the potential of our method and motivate the use causality-driven methods for forecasting.

## 1 Introduction

Deep learning has achieved profound success in vision and language modelling tasks (Brown et al., 2020; Nichol et al., 2021). Still, it remains a grand challenge and a prominent research direction to enable deep neural networks to perform causal discovery and reasoning (Yi et al., 2020; Girdhar & Ramanan, 2020; Sauer & Geiger, 2021), which is an inherent mechanism in human cognition (Spelke & Kinzler, 2007). Specifically for analysing time series data, causal discovery involves identifying the underlying temporal causal structure of the observed sequences. Many existing causal discovery approaches for time series assume stationarity (Granger, 1969; Peters et al., 2017; Löwe et al., 2020; Li et al., 2020; Tank et al., 2021), which is restrictive as sequence data from real-world scenarios are often non-stationary with potential hidden confounders. Recent works introduce a number of different assumptions to tackle causal discovery for non-stationary time series (Zhang et al., 2017; Ghassami et al., 2018; Huang et al., 2019), but in general, causal discovery on non-stationary time series under mild and realistic assumptions is an open problem.

This work aims at addressing this open challenge by proposing a causal discovery algorithm for *condionally stationary time series*, for which the dynamics of the observed system change depending on a set of "state" variables. This assumption holds for many real-world scenarios, e.g., with people who behave differently and take different decisions depending on underlying factors such as mood, previous experience, and the actions of other agents. The causal discovery task for such conditionally stationary time series poses different challenges depending on the observability of the states, which is classified into 4 different scenarios:

1. **Scenario class 1** concerns the simplest case, where the states are observed and their dynamics are independent on other observed time series data (Figure 1a).

2. In **Scenario class 2**, the states are unobserved and directly dependent on observed variables. Figure 1b shows an example, where the states of the variables change according to their positions (pink vs purple regions). Another example is to consider an agent moving in a room where different behaviors are observed depending on their location.

3. **Scenario class 3** is more challenging: the state depends on earlier events, and thus cannot be directly inferred from observations. E.g., in Figure 1c, particles that change state upon collision. Also in a football game a player acts differently depending on earlier actions of the others.

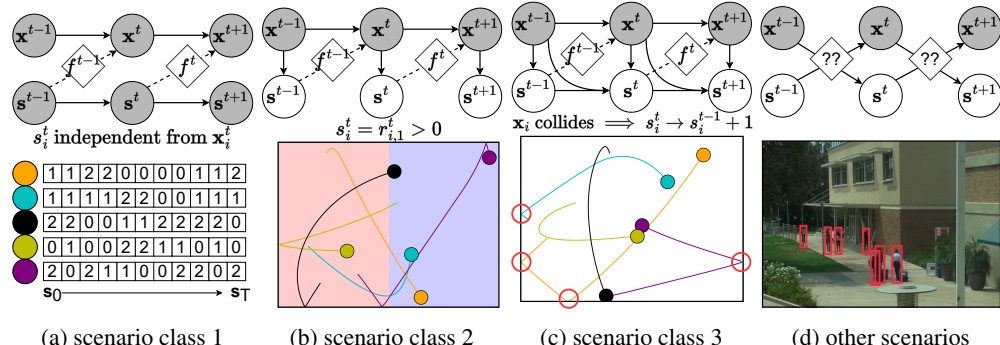

Figure 1: Graphical representations of the data generation processes considered in this work. $\mathbf{x}^t$ represents the observations of a time series sequence, and $\mathbf{s}^t$ denotes the state variables. The state affects the future observations by changing the causal structure (denoted as $f^t$ for different state values). The representations are examples of (a) scenario class 1, (b) scenario class 2, (c) scenario class 3, and (d) other scenarios (image adapted from Oh et al. (2011)).

4. Finally, many real-world scenarios (e.g., Figure 1d) are governed by underlying states that are not fully identifiable from the observations over time. Here the states can be unknown confounders to the observed time series, thus the causal discovery task is ill-defined.

Our approach, named State-Dependent Causal Inference (SDCI), is based on discovering the summary graphs (Peters et al., 2017) *conditioned on states* given observed sequences. It fits a graph neural network based variational auto-encoder (Löwe et al., 2020) to the non-stationary time series data, which enables efficient amortization for causal discovery across multiple observation sequences. We prove identifiability results for cases with fully-observed states; empirically SDCI also applies to cases with hidden states, which is confirmed by experiments on both synthetic linear datasets and spring data (See Figures 1b & 1c), covering scenario classes 1-3. Compared to baselines including a non-causal RNN-based approach, SDCI achieves better accuracy in identifying the underlying causal graph and forecasting future trajectories from historical observations, on both simulated and real-world data such as particle interactions and player trajectories in NBA games.

## 2 RELATED WORK

Causal discovery aims to identify causal relationships over a set of variables from observational data (Glymour et al., 2019). *Constraint-based* methods rely on conditional independence tests to recover the underlying DAG structure of the data. Representative approaches include the PC algorithm (Spirtes et al., 2000) and Fast Causal Inference (FCI) (Spirtes, 2001), and their extension to the time series data (Entner & Hoyer, 2010; Runge, 2018).

*Score-based* methods, such as Greedy Equivalence Search (GES) (Chickering, 2002), define and optimize score functions of causal graphs to identify the underlying causal structure. Regarding time series data, these methods are reformulated as learning *Dynamic* Bayesian Networks (DBNs) from data (Murphy et al., 2002). A recent approach in this line is DYNOTEARS (Pamfil et al., 2020), which aims to estimate both instantaneous and time-lagged relationships between variables in a time series without performing combinatorial search in the space of all possible graphs.

*Functional causal model-based* methods represent the effect as a function of its direct causes and their independent immeasurable noise (Shimizu et al., 2006; Zhang & Hyvärinen, 2009; Peters et al., 2014; Glymour et al., 2019). For time series, these approaches fit a dynamic model, often with constrained functional forms and connection sparsity in favor of identifiability (Peters et al., 2013).

Our work is concerned in modelling non-stationary time series using state variables as entities responsible for changing the dynamics along the sequence. Most relevant to ours is Amortized Causal Discovery (ACD) (Löwe et al., 2020), which assumes stationary time series and amortizes summary graph extraction process from samples with different graphs but shared dynamics. Similar ideas are also proposed in Li et al. (2020) for video applications. We extend ACD by allowing the underlying causal structure to vary depending on some state variables. For other works, Huang et al. (2015) extended Gaussian Process regression to identify time-varying functional causal models; Zhang et al. (2017) used kernel embeddings to detect distribution shifts in heterogeneous data, and Ghassami

et al. (2018) & Huang et al. (2019) estimated the time-varying causal effects. However, the latter two methods are limited by their linear causal model assumptions and fixed causal structures. Saggioro et al. (2020) considered causal discovery for regime-dependent dynamics. Our approach is more generic as we allow each variable in the multi-variate time series to have its own states, while regimes can be viewed as "global states" applied to all the variables.

## 3 STATE-DEPENDENT CAUSAL INFERENCE (SDCI)

We introduce SDCI to extract causal graphs from time series data where their dynamics are altered by means of a set of categorical variables, referred to as their states.

### 3.1 PROBLEM FORMULATION

We consider a dataset $\mathcal{D}$, where each sample $\mathbf{X} \sim \mathcal{D}$ consists of $N$ non-stationary time series $\mathbf{X} = \{\mathbf{x}_1, \ldots, \mathbf{x}_N\}$ of length $T$. We denote element $i$ at time-step $t$ as $\mathbf{x}_i^t \in \mathbb{R}^d$.

**Stationary time series.** We assume that the data generation process obeys a structural causal model (SCM) (Pearl, 2009), where all the observed variables $\mathbf{x}_i^t \in \mathcal{V}^{1:T}$ are its vertices, for each time series $i$ at each time-step $t$, and that there are no instantaneous effects and no hidden confounders. Moreover, same as the definitions of Granger causality (Granger, 1969), we assume that edges of a causal graph cannot go back in time. For the sake of clarity, the underlying causal structure satisfies the first-order Markov property in this work; however, one could directly extend it to the higher order. When considering causality in time series, the corresponding causal graph of a SCM is called *full time graph* (Peters et al., 2017), $\mathcal{G}^{1:T}$. Note that our treatment of stationarity refers to the structure of the causal mechanisms, i.e., edges are invariant in time.

Assumptions similar to the previous ones have been introduced in the past for the analysis of causality considering stationary time series data (Löwe et al., 2020; Li et al., 2020), where the task is to extract the *summary graph*, $\mathcal{G} = \{\mathcal{V}, \mathcal{E}\}$, where $\mathcal{V} = \{\mathbf{x}_1, \ldots, \mathbf{x}_N\}$ and an edge from $\mathbf{x}_i$ to $\mathbf{x}_j$ is included in $\mathcal{E}$ if there exists $1 \leq t < t' \leq T$ such that there is an edge from $\mathbf{x}_i^t$ to $\mathbf{x}_j^{t'}$ in the full time graph. Both *full time graph* and *summary graph* are identifiable from observational data as shown by Peters et al. (2013) if the generative process satisfies *Time Series Models with Independent Noise* (TiMINo). We further assume first-order Markov property and an additive noise model (ANM) without instantaneous effects for the generative process:

$$\mathbf{x}_j^t = f_j\left((\mathbf{PA}_j^1)^{t-1}\right) + \boldsymbol{\epsilon}_j^t \tag{1}$$

where $\mathbf{PA}_j^1 \subseteq \mathcal{V}$ denotes the parents of $\mathbf{x}_j$ and $\boldsymbol{\epsilon}_j^t$ denotes independent noise. The direct causes of $\mathbf{x}_j^t$ are $\mathbf{PA}(\mathbf{x}_j^t) = (\mathbf{PA}_j^1)^{t-1} \subset \mathcal{V}^{1:T}$. As here the time series is stationary, we can regard this as first querying the summary graph $\mathcal{G}$ to extract the parents, and then using the correct temporal index $(t-1)$ as a superscript of each parent $(\mathbf{PA}_j^1)$.

**Conditionally stationary time series.** For non-stationary time series considered in this work, we assume that at each time-step $t$ we have access to state variables $\mathbf{s}^t = \{s_1^t, \ldots, s_N^t\}$, where $s_i^t \in \{1, \ldots, K\}$ controls the causal effects of $\mathbf{x}_i^t$ to future variables $\mathbf{x}_{1:N}^{t+1}$. In other words, when the state $s_i^t$ changes, so do the causal effects for $\mathbf{x}_i^t$. We further assume that the time series is stationary if the states are held constant along the sequence, i.e., $\mathbf{s}^1 = \cdots = \mathbf{s}^T$. For the general case where $\mathbf{s}^t$ changes through time, even though the time series is non-stationary, it is *conditionally stationary* given the state variables $\mathbf{S} = \{\mathbf{s}^1, \ldots, \mathbf{s}^T\}$.

Figure 2a illustrates the full time graph of a conditionally stationary time series example following our assumptions. In this case a variable affects different sets of future variables depending its state, and the time series is non-stationary as $\mathbf{s}^1 \neq \mathbf{s}^2 \neq \mathbf{s}^3$. Note that the states are not included explicitly in the SCM of the generative process (otherwise the SCM requires additional variables with their own vertices for $\mathbf{S}$). This is a valid simplification since we assume access to the states at all times. Our theoretical results presented below do not apply when considering hidden states, since one must then explicitly model its causal structure in the SCM.

**Conditional summary graph.** Our goal for causal discovery is to recover the *full time graph*, where for stationary time series this can be achieved by identifying the summary graph under the first-order Markov assumption. However, for conditionally stationary time series, the summary

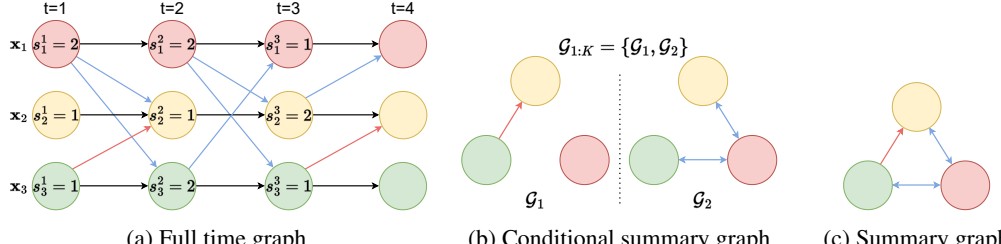

Figure 2: (a) Full time graph $\mathcal{G}^{1:T}$ of a sample considering our problem setting, (b) conditional summary graph $\mathcal{G}_{1:K}$, and (c) summary graph $\mathcal{G}$ of the corresponding sample. Different colors (red and blue) denote effects caused by different states.

graph can be non-informative (probably close to a fully connected graph), due to different causal effects induced by variables in different states. As an example, Figure 2c shows the summary graph extracted from the full time graph of Figure 2a. Being dense, the summary graph as defined for stationary time series is less useful in non-stationary settings. To address this issue, for conditionally stationary time series we define the *conditional summary graph*.

**Definition 1** (Conditional summary graph, first-order Markov setting). *Given a full time graph $\mathcal{G}^{1:T}$, its conditional summary graph is a set of $K$ summary graphs, $\mathcal{G}_{1:K} = \{\mathcal{G}_k : 1 \le k \le K\}$, where $K$ is the number of possible state values. Each summary graph $\mathcal{G}_k = \{\mathcal{V}, \mathcal{E}_k\}$ has the same vertices $\mathcal{V} = \{\boldsymbol{x}_1, \dots, \boldsymbol{x}_N\}$, and an edge from $\boldsymbol{x}_i$ to $\boldsymbol{x}_j$ is added to $\mathcal{E}_k$ if there exists a time-step $1 \le t \le T$ such that $s_i^t = k$ and in $\mathcal{G}^{1:T}$, $\boldsymbol{x}_i^t$ is connected to $\boldsymbol{x}_j^{t+1}$.*

Note that both summary and conditional summary graphs do not include auto-regressive connections $\mathbf{x}_i^t \to \mathbf{x}_i^{t+1}$. Figure 2b shows the conditional summary graph extracted from the full time graph of Figure 2a. For $k = 1$ we have $s_3^1 = 1$ and there is a "red edge" connecting $\mathbf{x}_3^1$ and $\mathbf{x}_2^2$, therefore for $\mathcal{G}_1$ there is a "red edge" in the edge set $\mathcal{E}_1$. Similar reasoning applies for $\mathcal{G}_2$. Compared to the summary graph 2c, the conditional summary graph contains a compact, yet valid representation of the causal structure, and thus being more informative. Conditional summary graph is more widely applicable beyond regime-dependent dynamics (see e.g., (Saggioro et al., 2020)): we do not restrict the entire time series to be in one of the $K$ states, but allow each variable have its own states.

**State-dependent TiMINo.** We extend TiMINo to conditionally stationary time series with observed states and show the identifiability of both full time graph and conditional summary graph. Consider the update on $\mathbf{x}_j^t$ as in Eq. 1. First for non-stationary time series the direct causes of $\mathbf{x}_j^t$, $\mathbf{PA}(\mathbf{x}_j^t)$, are no longer constant in time. However, for conditionally stationary time series satisfying first-order Markov property and our assumptions specified above, $\mathbf{PA}(\mathbf{x}_j^t)$ is determined by the states at the previous time-step $t-1$. Thus we can write $\mathbf{PA}(\mathbf{x}_j^t) = (\mathbf{PA}_j^1|\mathbf{s}^{t-1})^{t-1}$ and the *state-dependent TiMINo* as (with first-order Markov assumption, an ANM model and no instantaneous effect):

$$\mathbf{x}_j^t = f_j^{\mathbf{s}^{t-1}}\left((\mathbf{PA}_j^1|\mathbf{s}^{t-1})^{t-1}\right) + \boldsymbol{\epsilon}_j^t, \quad \mathbf{PA}_j^1|\mathbf{s}^{t-1} = \{\mathbf{x}_i : \mathbf{x}_j \in C_i(s_i^{t-1}), 1 \le i \le N\}, \quad (2)$$

where $s_i^{t-1} \in \{1, \dots, K\}$ and $C_i(k) \subseteq \mathcal{V}$ denotes the children of variable $\mathbf{x}_i$ when its associated state equals to $k$. To illustrate, in Figure 2a we have that $\mathbf{PA}(\mathbf{x}_2^2) = \{\mathbf{x}_1^1, \mathbf{x}_2^1, \mathbf{x}_3^1\}$ because $\mathbf{x}_2 \in C_1(s_1^1)$ and $\mathbf{x}_2 \in C_3(s_3^1)$; however $\mathbf{PA}(\mathbf{x}_2^3) = \{\mathbf{x}_1^2, \mathbf{x}_2^2\}$ because now $\mathbf{x}_2 \in C_1(s_1^2)$ but $\mathbf{x}_2 \notin C_3(s_3^2)$.

**Identifiability of state-dependent TiMINo.** As we assume access to the state variables at all times, the original properties of Markov assumption and causal minimality are maintained, which are satisfied by TiMINo. Consequently, the *conditional summary graph* becomes identifiable by extending the indentifiability proof of Peters et al. (2013) to our setting and further assuming that all the states are visited at least once. Notice that failing to observe data corresponding to a particular state of one variable makes it practically impossible to capture the underlying effect of that variable conditioned on the unvisited state. More details for the preservation of causal minimality and Markov condition in the state-dependent TiMINo, the identifiability of both the full time graph and conditional summary graph, and extensions to incorporate other time-lagged or instantaneous effects can be found in Appendix A and B.

Considering the previous results, one could directly extend *TiMINo causality* (Peters et al., 2013) to conditionally stationary time series. The algorithm uses a fitting method (to estimate $f_j$ in Eq.1)

and tests for independence using residuals. We notice that in our setting, the direct causes of $\mathbf{x}_j^t$ depend on the state values $\mathbf{s}^{t-1}$, which represent $K^N$ models in total ($f_j^{\mathbf{s}^{t-1}}$). This renders such extension impractical for efficient causal discovery in such scenarios. However, our key observation is that in real-world non-stationary data, many of the underlying function components are shared not only across data but also across variables, i.e., $f_i^{\mathbf{k}} = f_j^{\mathbf{l}}$ for some variables $i, j$ and state sets $\mathbf{k}, \mathbf{l}$. This benefits the use of deep learning as effectively more data is available to identify the individual effects (where the total number of them is significantly less than $K^N$). Therefore, we propose a deep learning-based approach in a similar spirit as Löwe et al. (2020), to amortise the discovery of causal relationships across data samples within the same dataset. We leave the proof of our method regarding consistent parameter estimates (Geffner et al., 2022) to future work.

**State-dependent causal inference.** We mainly focus on non-stationary causal graphs which may have different edge-types at different times. Based on our assumptions, the interaction (i.e., edge-type) $\mathbf{x}_i \rightarrow \mathbf{x}_j$ can change according to the state of the variable $\mathbf{x}_i$. Following Kipf et al. (2018); Li et al. (2020); Löwe et al. (2020), we consider this edge-type for $\mathbf{x}_i \rightarrow \mathbf{x}_j$ at time $t$ as a categorical variable $z_{ij}^t \in \{0, \ldots, n_\epsilon - 1\}$ which can represent $n_\epsilon$ interaction types between pairs of variables. Specifically we use edge-type 0 to denote "no causal effect" between two variables. Notice that the edge-type differs from the state: the former controls the functional form of the causal effect and the latter allows the variables to affect others differently along time. For example, one can have $\mathbf{x}_i \rightarrow \mathbf{x}_j$ whenever $z_{ij}^t \neq 0$, but the functional form of the causal relationship can differ.

Our method focuses on extracting a *conditional summary graph* $\mathcal{G}_{1:K}$ (assuming $K$ states). Previous approaches aiming for this task assume stationary time series data for which $\mathcal{G}_1 = \cdots = \mathcal{G}_K$. For conditionally stationary time series, we extract the $k$-th summary graph including edge-types, $\tilde{\mathcal{G}}_k = \{\mathcal{V}, \tilde{\mathcal{E}}_k\}$, where $\tilde{\mathcal{E}}_k$ and $\tilde{\mathcal{E}}_{k'}$ can differ for $k \neq k'$. We further define $\tilde{\mathcal{E}}_k = \{w_{ijk} \in \{0, \ldots, n_\epsilon - 1\} : \mathbf{x}_i, \mathbf{x}_j \in \mathcal{V}\}$ as the collection of edge types for variable pairs in $\mathcal{V}$. Note here that an edge is visualised in the conditional summary graph visualisation only when $w_{ijk} \neq 0$ (see Figure 2b). Then the edge-type interaction $\mathbf{x}_i \rightarrow \mathbf{x}_j$ can be queried at each time-step $t$ as follows:

$$z_{ij}^t = \left(\tilde{\mathcal{E}}_{s_i^t}\right)_{ij}, \qquad \left(\tilde{\mathcal{E}}_{s_i^t}\right)_{ij} = w_{ijk} \text{ if } s_i^t = k. \qquad (3)$$

Therefore, causal discovery for conditionally stationary time series requires extracting the (unknown) conditional summary graph given observations of $\mathbf{X}$ (and perhaps also $\mathbf{S}$). It not only requires to design a parametrizable function to infer the causal structure, but also to evaluate how this inference fits to the input observations.

## 3.2 Implementation

We introduce a probabilistic approach which models the distribution of the edge-types $\{z_{ij}^t\}$ given observed data. Following our assumptions for conditionally stationary time series, this task can be solved by learning the distribution of $\mathbf{W} = \{w_{ijk} : 1 \leq i, j \leq N, 1 \leq k \leq K\}$ given data, as after inferring $\mathbf{W}$ one can then query the edge-types $z_{ij}^t$ as in Eq. 3. Inspired by previous approaches (Li et al., 2020; Löwe et al., 2020; Kipf et al., 2018), our implementation is based on a variational auto-encoder (VAE) (Kingma & Welling, 2014) and graph neural networks. We first discuss our approach for the case with fully observed states, then extend the method to the hidden state regime. A diagram of the proposed approach is visualised in Figure 3.

**Encoder.** Similar to Löwe et al. (2020), we use a factorized $q$ distribution $q_\phi(\mathbf{W}|\mathbf{X}, \mathbf{S}) = \prod_{k=1}^K \prod_{ij} q_\phi(w_{ijk}|\mathbf{X}, \mathbf{S})$. The encoder receives both $\mathbf{X}$ and $\mathbf{S}$ as the input, and extracts an embedding that represents the causal interaction conditioning on the state for every possible edge $\mathbf{x}_i \rightarrow \mathbf{x}_j$. We compute approximate posterior $q_\phi(w_{ijk}|\mathbf{X}, \mathbf{S})$ as follows,

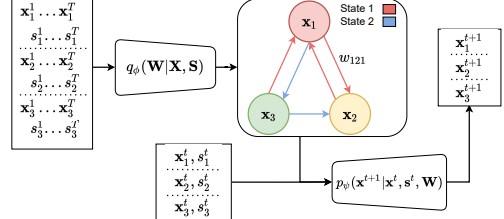

$$\phi_{ij} = f_\phi(\mathbf{X}, \mathbf{S})_{ij} \in \mathbb{R}^{K \times n_\epsilon}, \qquad (4)$$
$$q_\phi(w_{ijk}|\mathbf{X}, \mathbf{S}) = \Theta(\phi_{ijk}/\tau), \qquad (5)$$

Figure 3: SDCI extracts a *conditional summary graph* that describes the edge-type interaction for every pair of edges conditioned on the states.

where $f_\phi(\mathbf{X}, \mathbf{S})$ is a neural network that returns outputs $\{\phi_{ij} : 1 \leq i, j \leq N\}$ given the input as $\mathbf{X}$ concatenated with a one-hot representation of the state variable $\mathbf{S}$. $\Theta(\cdot)$ denotes a softmax activation

with temperature $\tau$. Note here the softmax activation is taken over the $k$-th row vector $\phi_{ijk} \in \mathbb{R}^{n_\epsilon}$ in $\phi_{ij}$, which is the edge embedding for $\mathbf{x}_i \to_{ij} \mathbf{x}_j$ at state $k$. As this returns a categorical distribution for $q_\phi(w_{ijk}|\mathbf{X}, \mathbf{S})$, we apply the Gumble-softmax trick (Maddison et al., 2017; Jang et al., 2016) to enable direct differentiation for back-propagation during training.

The construction of $f_\phi(\mathbf{X}, \mathbf{S})$ is based on graph neural networks and MLPs. See Appendix C.2 for more details.

**Decoder for X.** The decoder for the observations $\mathbf{X}$ given edge-types $\mathbf{W}$ (e.g., sampled as $w_{ijk} \sim q_\phi(w_{ijk}|\mathbf{X}, \mathbf{S})$) and the states $\mathbf{S}$ are defined following the first-order Markov assumption (with $\mathbf{x}^0, \mathbf{s}^0$ as dummy variables):

$$p_\psi(\mathbf{X}|\mathbf{W}, \mathbf{S}) = \prod_{t=0}^{T-1} \prod_{j=1}^{N} p_\psi(\mathbf{x}_j^{t+1}|\mathbf{x}^t, \mathbf{s}^t, \mathbf{W}) = \prod_{t=0}^{T-1} \prod_{j=1}^{N} \mathcal{N}(\tilde{\mathbf{x}}_j^{t+1}, \sigma^2 \mathbb{I}). \tag{6}$$

The mean $\tilde{\mathbf{x}}_j^{t+1}$ is defined as follows. Given sampled edge-types $\mathbf{W} = \{w_{ijk}\}$, the decoder first queries the edge-type for element $j$ at time $t+1$ as $z_{ij}^t = w_{ijk'}$ for $s_i^t = k'$ (also see Eq. 3). Then the information along the predicted edge-type interactions is retrieved and aggregated as follows [1]

$$\mathbf{h}_{ij}^t = \sum_{e>0} \mathbf{1}_{(z_{ij}^t=e)} f_e(\mathbf{x}_i^t, \mathbf{x}_j^t), \quad \tilde{\mathbf{x}}_j^{t+1} = \mathbf{x}_j^t + f_p\Big(\sum_{i \neq j} \mathbf{h}_{ij}^t, \mathbf{x}_j^t\Big), \tag{7}$$

where $\{f_e\}_{e=1}^{n_\epsilon-1}$ is a set of parametrizable functions, one defined for each edge type excluding the no-edge interaction. $f_p$ is a neural network that aggregates the information from the previous time-step, and models the dynamics for each variable $\mathbf{x}_j^{t+1}$.

**Objective.** The encoder $q_\phi(\mathbf{W}|\mathbf{X}, \mathbf{S})$ and the decoder & state dynamic model $p_\psi(\mathbf{X}, \mathbf{S}|\mathbf{W})$ are trained using a modified VAE objective:

$$\mathcal{L} = \sum_{t=0}^{T-1} \mathbb{E}_{q_\phi(\mathbf{W}|\mathbf{X},\mathbf{S})} \big[ \log p_\psi(\mathbf{x}^{t+1}|\mathbf{x}^t, \mathbf{s}^t, \mathbf{W}) \big] - KL\big(q_\phi(\mathbf{W}|\mathbf{X}, \mathbf{S})||p(\mathbf{W})\big) \tag{8}$$

where we use $\lambda$ to balance the accuracy in learning the decoder for $\mathbf{X}$ and the state dynamics. Here we used a factorised prior $p(\mathbf{W}) = \prod_{k=1}^{K} \prod_{ij} p_k(w_{ijk})$ which acts as a regularizer over the inferred edge-type distribution $q$. In our experiments we set this prior to be Uniform$(\{0, \ldots, n_\epsilon - 1\})$, although in certain applications it may be useful to use different $p_k$ to encourage different sparsity levels for $\mathcal{G}_k$.

**Hidden state regime.** Often in practice only observations of $\mathbf{X}$ are available, i.e., the states $\mathbf{S}$ are hidden variables. In this case we make a factorised approximation $q_\phi(\mathbf{W}, \mathbf{S}|\mathbf{X}) = q_\phi(\mathbf{W}|\mathbf{X})q_\phi(\mathbf{S}|\mathbf{X})$, where $q_\phi(\mathbf{W}|\mathbf{X})$ is defined in a similar way as in the fully-observed case (see above) except for using network $f_\phi(\mathbf{X})$ for the softmax logits. For the states, we consider $q_\phi(\mathbf{S}|\mathbf{X}) = \prod_{t=1}^{T} \prod_{i=1}^{N} q_\phi(s_i^t|\mathbf{x}_i^t)$, with each $q_\phi(s_i^t|\mathbf{x}_i^t)$ defined as

$$q_\phi(s_i^t|\mathbf{x}_i^t) = \Theta(\hat{s}_i^t/\gamma), \quad \hat{s}_i^t = \hat{f}_s(\mathbf{x}_i^t), \tag{9}$$

with $\gamma < 1$ as a temperature factor and $\hat{f}_s$ as a neural network. For the decoding process given $\mathbf{W} \sim q_\phi(\mathbf{W}|\mathbf{X})$, we also sample states $\mathbf{S} \sim q_\phi(\mathbf{S}|\mathbf{X})$ using the categorical reparameterization (Maddison et al., 2017; Jang et al., 2016). Therefore, the decoding step can be performed as in Eq. 7 and resulting message passing operation becomes

$$\mathbf{h}_{ij}^t = \sum_{k=1}^{K} \mathbf{1}_{(s_i^t=k)} \sum_{e>0} \mathbf{1}_{(w_{ijk}=e)} f_e(\mathbf{x}_i^t, \mathbf{x}_j^t), \tag{10}$$

and we apply similar Gumbel-softmax tricks as in the fully-observed state case to replace $\mathbf{1}_{(s_i^t=k)}$ and $\mathbf{1}_{(w_{ijk}=e)}$ during training. The VAE training objective in this case is as follows

$$\mathcal{L} = \sum_{t=0}^{T-1} \mathbb{E}_{q_\phi(\mathbf{W},\mathbf{S}|\mathbf{X})} \big[ \log p_\psi(\mathbf{x}^{t+1}|\mathbf{x}^t, \mathbf{s}^t, \mathbf{W}) \big] - KL\big(q_\phi(\mathbf{W}|\mathbf{X})||p(\mathbf{W})\big) \tag{11}$$

where we set $q_\phi(\mathbf{S}|\mathbf{X}) = p(\mathbf{S}|\mathbf{X})$, and thus $KL\big(q_\phi(\mathbf{S}|\mathbf{X})||p(\mathbf{S}|\mathbf{X})\big) = 0$. Note that the objective is still a valid lower bound of $p(\mathbf{X})$. See Appendix C.1 for an alternate formulation. In this hidden state regime the previous theoretical guarantees on identifiability no longer hold due to violations of our assumptions. However, this setting can still be useful as a showcase for analysis of non-stationary time series and possible directions of future work.

---

[1] In training, $\mathbf{1}_{(z_{ij}^t=e)}$ is replaced by the $e$-th dimension of $w_{ijk'}$ which is sampled from $q_\phi$ with Gumbel-softmax relaxation.

## 4 EXPERIMENTS

We evaluate SDCI on two synthetic non-stationary time series datasets, and realistic data based on NBA in-game player movements. Our results are compared to ACD (Löwe et al., 2020), TdCM Huang et al. (2015), CD-NOD (Zhang et al., 2017), SAEM (Huang et al., 2019), and a non-causal deep learning method called variational RNN (VRNN) (Chung et al., 2015).

### 4.1 EXPERIMENTS ON LINEAR DATA

We start from linear message passing operations between different time series. Conditioned on the states, the variables $\mathbf{x}_i^t, \mathbf{x}_j^{t+1} \in \mathbb{R}$ are connected by an edge of $n_\epsilon$ different types. Each edge-type is captured by the linear coefficients $\{\beta_k \in \mathbb{R}\}_{k=0}^{n_\epsilon-1}$ with the convention that $\beta_0 = 0$ (no connection). Notice that the effect of $\mathbf{x}_i^t$ on $\mathbf{x}_j^{t+1}$ changes during time according to the state value $s_i^t$ and the underlying conditional summary graph $\mathcal{G}_{1:K}$. We report experiments on scenario class 1 in Appendix F.1 and study scenario class 2 below. Details on data generation, training settings, and additional experiments can be found in Appendices D.1, C.3, and F.1 respectively.

**Results for scenario class 2.** Scenario class 2 considers unobserved states (only $\mathbf{X}$ is given, see Figure 1b), enabling comparisons with CD-NOD, SAEM and TdCM which perform causal discovery based on $\mathbf{X}$ only. As these baselines consider constant causal connections (i.e., a single summary graph), for a fair comparison we evaluate the identification accuracy of the summary graph such that it only considers the existence of an edge rather than capturing the edge type. We simulate this setting with 2 edge-types (2-EDGE), 3 edge-types with constant summary graphs (3-EDGE CONST) and 3 edge-types with no constraints on the conditional summary graph (3-EDGE FREE).

Table 1: Summary graph and conditional summary graph accuracy for linear data in scenario class 2.

| METHOD | SG ACCURACY | | |
| --- | --- | --- | --- |
| | 2-EDGE | 3-EDGE | |
| | | CONST | FREE |
| TDCM (T=100) | $65.17 \pm 2.65$ | $63.67 \pm 1.61$ | $63.50 \pm 1.62$ |
| CD-NOD (T=100) | $39.33 \pm 2.59$ | $35.25 \pm 2.51$ | $28.58 \pm 2.66$ |
| SAEM (T=100) | $47.75 \pm 3.67$ | $39.04 \pm 2.38$ | $51.44 \pm 3.81$ |
| TDCM (T=1000) | $68.25 \pm 2.29$ | $61.17 \pm 2.28$ | $62.00 \pm 2.14$ |
| CD-NOD (T=1000) | $50.08 \pm 2.59$ | $42.08 \pm 2.17$ | $41.58 \pm 2.02$ |
| SAEM (T=1000) | $47.38 \pm 4.10$ | $25.93 \pm 2.82$ | $28.49 \pm 3.28$ |
| ACD (T=50) | $60.45 \pm 1.60$ | $87.00 \pm 2.56$ | $49.25 \pm 3.05$ |
| SDCI (T=50) | $97.08 \pm 1.05$ | $90.17 \pm 2.22$ | $64.00 \pm 2.93$ |
| | CSG ACCURACY | | |
| | 2-EDGE | 3-EDGE | |
| | | CONST | FREE |
| SDCI (T=50) | $98.08 \pm 0.64$ | $76.04 \pm 2.05$ | $65.45 \pm 1.99$ |

We report the (conditional) summary graph identification accuracies in Table 1. It is clear that SDCI performs the best in extracting both the summary graph and conditional summary graph. Since ACD assumes stationary time series, it achieves comparable results when considering a constant causal graph. On the other hand, TdCM, CD-NOD and SAEM perform significantly worse, as they are designed for non-stationary time series with distribution shifts and smooth time-varying coefficients, which are not suited for modeling discrete changes in dynamics. Furthermore, they conduct causal discovery on each multivariate time series separately, while ACD and SCDI benefit from amortized causal graph inference that utilises shared information across different time series.

### 4.2 EXPERIMENTS ON NONLINEAR SPRING DATA

We evaluate SCDI on *spring data* adapted from Kipf et al. (2018); Löwe et al. (2020), which consists of particles (or balls) connected by springs with directed impact - meaning that e.g. particle $i$ could affect particle $j$ with a force through a connecting spring, but leaving particle $i$ unaffected by this spring force. We consider 2 edge-types (presence/absence of directed spring) and scenario classes 1 and 2. We report results with scenario class 3 in Appendix F.2 Details of data generation and model hyper-parameters can be found in Appendices D.2 and C.3 respectively.

**Results for scenario class 1.** In this experiment the states are known and independent from the observations. For the ground truth dynamics, the state transitions incrementally into the next one every 10 time-steps. We report edge identification accuracy, which regards the conditional summary graph including edge-types ($\tilde{\mathcal{G}}_{1:K}$). Figure 4a shows results with increasing variables and increasing states, where we compare with ACD (dashed lines) as a stationary baseline. Although performance drops as $K$ increases, SDCI is able to maintain reasonable accuracy in edge-type identification in comparison to the stationary baseline. With increasing number of variables both approaches see accuracy drops; our hypothesis is that this can be addressed by increasing model capacity. The next

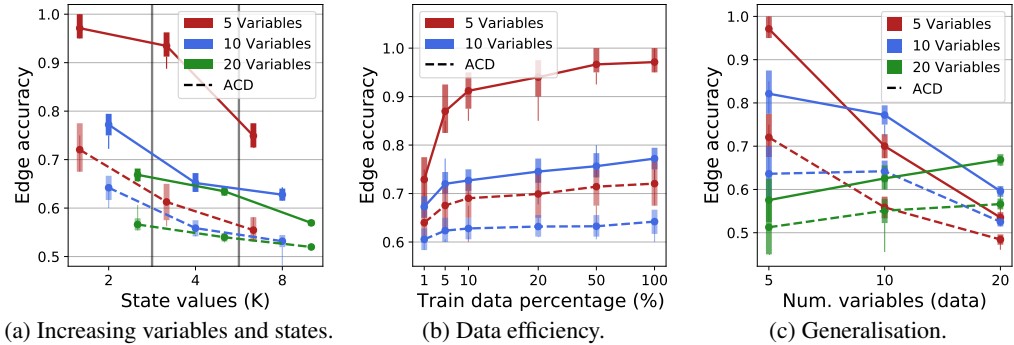

Figure 4: Results on spring data with scenario class 1 for (a) increasing variables and state values, (b) data efficiency, and (c) generalisation, where the x-axis indicates the variables in the test data and the legend indicates the model used. SDCI results are shown with solid lines.

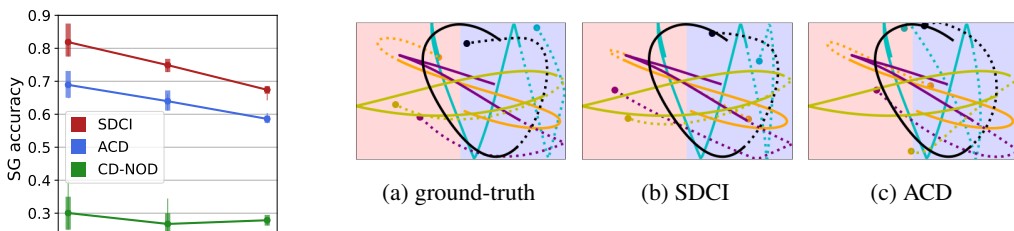

Figure 5: Results on spring data with scenario class 2.

Figure 6: Time series forecasting (dotted lines) of both (b) SCDI and (c) ACD for 50 time-steps along with (a) the ground-truth. The trajectories in solid lines are used as the input and the background color represents the state value.

test considers data efficiency of SDCI with results reported in Figure 4b. We see that both SDCI and ACD are data efficient in this scenario, where training on 10% of the data returns good performance already. Finally, we also report in Figure 4c on how these methods generalise to unseen data with different number of variables. Here both methods generalise better to settings where the number of variables is similar to the ones they were trained.

**Results for scenario class 2.** We consider the hidden state regime, where the underlying state of a particle changes depending on its location in the box ($K = 2$, see Figure 1b). Results in Figure 5 show a clear advantage of SDCI over baselines in terms of summary graph identification accuracy. Again, ACD is limited by its stationary dynamics assumption, and CD-NOD fails due to its inability in handling discrete changes in the causal effects of the full time graph. We visualize the 50 time-steps forecasts with SDCI and ACD given an input sequence of $T = 80$ time-steps in Figure 6. The conditional stationary assumption of SDCI allows it to extract more accurate graph structures, which results in better predictions than ACD. Overall, SDCI successfully decomposes the non-stationary dynamics into the conditional stationary ones while capturing the state transition dynamics.

### 4.3 EXPERIMENTS ON NBA PLAYER TRAJECTORIES

The last experiment considers modeling NBA player movements (Linou, 2016), which is a real-world multi-agent trajectory dataset with highly non-linear & non-stationary dynamics. Details of this dataset can be found in Appendix D.3, including our design of states (as ground-truth is unavailable), which is dependent on the player positions in the court. We evaluate SDCI in both scenarios (states observed/unobserved). To simplify the task, we only model the trajectories of the players (position and velocity), and condition the predictions on the position and velocity of the ball, by modifying Eq. 7 to include the ball features in the message passing aggregation of the decoder. Apart from ACD, we further include a non-causal baseline VRNN (Chung et al., 2015), which is a popular deep learning approach for forecasting. All the models are trained on sequences of length $T = 100$ (see Appendix C.4 for VRNN training details).

Figure 7a shows forecast error (in MSE) of the player positions (without velocities) for $T = 100$ steps computed on a held out dataset. In general, the causal discovery methods benefit from using a

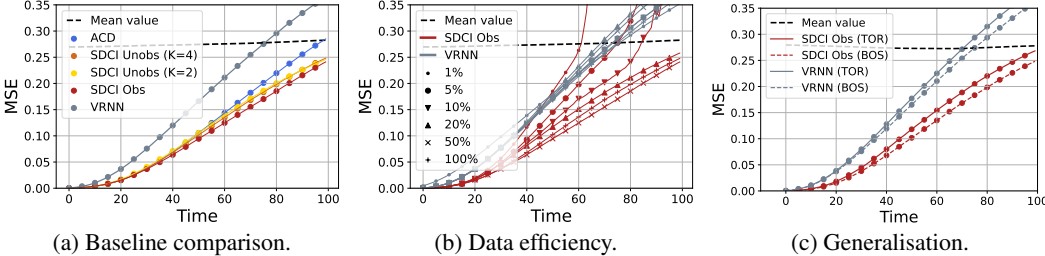

(a) Baseline comparison.    (b) Data efficiency.    (c) Generalisation.

Figure 7: Results on NBA data for (a) forecasting on player positions on test data, (b) data efficiency (% indicates the proportion of train data used), and (c) generalisation between NBA team matches. The dashed black line represents a baseline that always predicts the mean value over trajectories.

graph-based approach and outperform VRNN. In addition, SDCI produces more accurate long-term forecasts than ACD which is restricted by the stationarity assumption. For the hidden state setting, we report forecasts using $2$ and $4$ states, and observe comparable results to the observed case.

We evaluate and report in Figure 7b the data efficiency of SDCI in realistic settings. Again SDCI outperforms VRNN even in low-data regime. But more interestingly, SDCI better exploits the increasing amount of data as it continues to improve, especially on long-term prediction. These observations are in line with data efficiency test results in springs data (see Figure 4b).

Lastly we explore the generalisation ability of SDCI and evaluate the transferability of learned dynamics between different teams. Figure 7c shows forecast error for *Boston Celtics* matches, where the models are trained either on the same data (dashed line) or on *Toronto Raptors* matches (solid line). Both methods produce reasonable out-of-distribution forecasts, suggesting that they are able to learn shared dynamics between samples, and thus allowing generalisation on unseen data. Still SDCI outperforms VRNN considerably, showing its strong competence in forecasting for multi-agent scenarios with dynamical changes along the sequences.

To better understand the learned states from SDCI in the unobserved states scenario, we visualise in Figure 8 the state posterior as a function of the player's position $\mathbf{x}_i^t$ in the court. For 2 states, we observe that the state changes in regions close to the mid-court line. This is consistent with typical strategies in an NBA match,

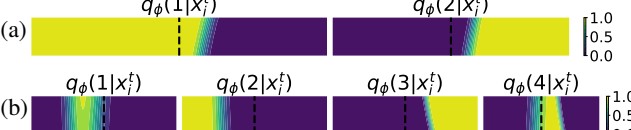

Figure 8: Learned regimes from SDCI on the NBA dataset using (a) $K = 2$ and (b) $K = 4$ state values. The dashed line indicates the mid court line, and the colour maps refer to the state posterior distribution $q_\phi(k|\mathbf{x}_i^t), k = \{1, \ldots, K\}$.

where the players first run to one side of the court, and then change their behaviour from "defence" to "offence" mode, or vice-versa. Similar arguments can be made for the 4-state case, where the court is segmented such that the second and third region boundaries are close to the 3-point line. Therefore, in addition to better forecast results, SDCI further provides better interpretability of the learned dynamics from data, and returns great promises in applications to other realistic scenarios.

## 5 CONCLUSIONS

We have developed SDCI for amortized causal discovery for conditionally stationary time series. Key to our development is the state-dependent TiMINo as an extension of Peters et al. (2013) to conditionally stationary time series, the new concept of conditional summary graph, and their identifiability under observed states. Evaluations on synthetic linear and nonlinear data show SDCI's improved accuracy in extracting the underlying causal graph and forecasting. Importantly, the improvement of SDCI over VRNN on modeling NBA player movements demonstrate the promise of causality-driven methods for forecasting and data interpretability.

A number of research directions are to be explored in the future. For theoretical studies, identifiability for hidden states will be derived with additional assumptions. For practical applications, SDCI will be scaled up and extended to video understanding, where e.g. the interacting objects are partially and noisily observed as semantically segmented regions. This would enable efficient and effective use of neural networks for causal reasoning in challenging real-world scenarios.

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

## A  IDENTIFIABILITY PROOFS

This section presents the detailed proofs for the theoretical results of our paper. Again we note here that the results only apply when the states **S** are fully observed. Our theory extend on the theoretical results presented in Peters et al. (2013) and Peters et al. (2017), which are presented first before our proofs.

**Proposition 1** (Prop. 6.36 in Peters et al. (2017)). *Causal minimality is satisfied if and only if* $\forall \boldsymbol{x}_j, \forall \boldsymbol{y} \in \boldsymbol{PA}(\boldsymbol{x}_j)$ *we have that* $\boldsymbol{x}_j \not\perp\!\!\!\perp \boldsymbol{y}|\boldsymbol{PA}(\boldsymbol{x}_j)\backslash\{\boldsymbol{y}\}$

**Proposition 2** (Prop. 7.4 in Peters et al. (2017)). *Consider a distribution given by the following additive noise model*

$$\boldsymbol{x}_j = f_j(\boldsymbol{PA}(\boldsymbol{x}_j)) + \boldsymbol{\epsilon}_j, \quad j = 1, \ldots, N \tag{12}$$

*if the functions $f_j$ are not constant in any of their arguments, then the joint distribution satisfies causal minimality with respect to the corresponding graph.*

Recall from the main text that *state-dependent TiMINo* is defined as (with first-order Markov assumption, additive noise model, and no instantaneous effect):

$$\mathbf{x}_j^t = f_j\left((\mathbf{PA}_j^1|\mathbf{s}^{t-1})^{t-1}\right) + \boldsymbol{\epsilon}_i^t, \tag{13}$$

$$\mathbf{PA}_j^1|\mathbf{s}^{t-1} = \{\mathbf{x}_i : \mathbf{x}_j \in C_i(s_i^{t-1}), 1 \le i \le N\}. \tag{14}$$

Below we derive the identifiability results for state-dependent TiMINo given fully observed states **S**.

### A.1  STATE-DEPENDENT TiMINo PRESERVES MARKOV CONDITION

We show that the state-dependent TiMINo is Markov.

**Lemma 1.** *Given the states S, if X is generated by a state-dependent TiMINo, then each variable is conditionally independent of each non-descendants given its parents.*

*Proof.* As the states **S** are given, we can retrieve the direct causes of $\mathbf{x}_j^t$:

$$P := \mathbf{PA}(\mathbf{x}_j^t) = \{\mathbf{x}_i^{t-1} : \mathbf{x}_j \in C_i(s_i^{t-1}), 1 \le i \le N\},$$

and use state-dependent TiMINo to compute $\mathbf{x}_j^t$ given its parents:

$$\mathbf{x}_j^t|_{P=p} = f_j(p) + \boldsymbol{\epsilon}_i^t.$$

Given its parents P, variable $\mathbf{x}_j^t$ is therefore independent of its non-descendants.  □

### A.2  STATE-DEPENDENT TiMINo PRESERVES CAUSAL MINIMALITY

We show that the state-dependent TiMINo model satisfies causal minimality given observed states.

**Lemma 2.** *Assume all the functions $f_j$ in a state-dependent TiMINo are not constant in any of their arguments. Then causal minimality is preserved given observed states.*

*Proof.* Again as the states **S** are observed, then for any $\mathbf{x}_j^t$ the direct causes of it can be retrieved by:

$$P := \mathbf{PA}(\mathbf{x}_j^t) = \{\mathbf{x}_i^{t-1} : \mathbf{x}_j \in C_i(s_i^{t-1}), 1 \le i \le N\}.$$

Assume causal minimality is not satisfied, then following Proposition 1, there exists $\mathbf{x}_j^t$ and $\mathbf{x}_i^{t-1} \in P$ such that

$$\mathbf{x}_j^t \perp\!\!\!\perp \mathbf{x}_i^{t-1}|P\backslash\{\mathbf{x}_i^{t-1}\}$$

Denote $P\backslash\{\mathbf{x}_i^{t-1}\}$ as $P_A$. Then, if $\mathbb{E}[\boldsymbol{\epsilon}_i^t] = 0$, there exists a function $c(\cdot)$ such that

$$\mathbb{E}\left[\mathbf{x}_j^t\right] = f_j\left(P_A, \mathbf{x}_i^{t-1}\right) = c(P_A), \quad \forall \mathbf{x}_i^{t-1}.$$

This implies that $f_j$ is constant with respect to $\mathbf{x}_i^{t-1}$. It contradicts with the assumption that $f_j$ is not constant in any of its arguments. Therefore, causal minimality in state-dependent TiMINo is preserved given observed states.  □

### A.3 Identifiability in State-Dependent TiMINo

Using the Markov and causal minimality results, we derived the following identifiability result for state-dependent TiMINo, and the proof is inspired by Peters et al. (2013).

**Theorem 1.** *Consider the state-dependent TiMINo, where only first-order Markov interactions are present, and there is no instantaneous effects. Then the full time graph $\mathcal{G}^{1:T}$ is identifiable from the distribution of data given states.*

*Proof.* Conditioned on $\mathbf{S}$, assume that $\mathbf{x}_t$ can be computed from two state-dependent TiMINo with two different full time graphs, $\mathcal{G}_1^{1:T}$ and $\mathcal{G}_2^{1:T}$. Suppose there is an edge $\mathbf{x}_i^{t-1} \rightarrow \mathbf{x}_j^t$ which is in $\mathcal{G}_1^{1:T}$, but not in $\mathcal{G}_2^{1:T}$.

1. From $\mathcal{G}_2^{1:T}$ and the Markov condition (Lemma 1), we have $\mathbf{x}_j^t \perp\!\!\!\perp \mathbf{x}_i^{t-1} | \{\mathbf{x}_k^{t-1}, 1 \leq k \leq N, k \neq i\}$.

2. From $\mathcal{G}_1^{1:T}$ and causal minimality (Lemma 2), we have $\mathbf{x}_j^t \not\perp\!\!\!\perp \mathbf{x}_i^{t-1} | \{\mathbf{x}_k^{t-1}, 1 \leq k \leq N, k \neq i\}$.

Since we have a contradiction, the full time graphs $\mathcal{G}_1^{1:T}$ and $\mathcal{G}_2^{1:T}$ must be equal. Therefore, the full time graph is identifiable from the distribution of data given states. $\square$

Given the identifiability results, in theory, we can estimate the state-dependent TiMiNo in Eq. 2 with maximum likelihood (Zhang et al., 2015). In our implementation, we train the model with the modified VAE objective 8. The empirical results show that it infers causal structures correctly; however, it is nontrivial to prove that the identifiability holds with the modified VAE objective, which has been discussed in Geffner et al. (2022) as well and will be our future work.

### A.4 Identifiability of conditional summary graph

Assuming the full time graph has been extracted successfully, one can deduce the structure of the conditional summary graph by observing the edges of pairs of variables conditioned on the state variable which is the cause of the interaction. In particular, the precise mathematical statement for this identifiability result is as follows, assuming that all the states of each element have been visited at least once in order to condition on each possible state.

**Corollary 1.** *Consider the state-dependent TiMINo, where only first-order Markov interactions are present, and there is no instantaneous effects. Then the conditional summary graph $\mathcal{G}_{1:K}$ is identifiable given the states $\mathbf{S}$, if for any $1 \leq i \leq N$ and any $1 \leq k \leq K$ there exists $1 \leq t \leq T$ such that $s_i^t = k$.*

*Proof.* Notice that $\mathcal{G}_k = \{\mathcal{V}, \mathcal{E}_k\}$ with $\mathcal{V} = \{\mathbf{x}_1, \ldots, \mathbf{x}_N\}$, therefore identifying $\mathcal{G}_k$ is equivalent to identifying the edge set $\mathcal{E}_k$. From the assumptions, the full time graph $\mathcal{G}^{1:T}$ is identifiable (Theorem 1). Then for each variable $\mathbf{x}_j^{t+1}$ we can retrieve from $\mathcal{G}^{1:T}$ the parents $\mathbf{PA}(\mathbf{x}_j^{t+1})$. Then for each $\mathbf{x}_i^t \in \mathbf{PA}(\mathbf{x}_j^{t+1})$, we can query its state $s_i^t$ and add an edge $\mathbf{x}_i \rightarrow \mathbf{x}_j$ to the edge set $\mathcal{E}_k$ if $s_i^t = k$. Now for any $i, k$ consider the $i$-th node in the $k$-th summary graph $\mathcal{G}_k$, since we assume that there exists $1 \leq t \leq T$ such that $s_i^t = k$, this makes sure that the edges coming out of node $\mathbf{x}_i$ at state $k$ are added to $\mathcal{E}_k$. Therefore this procedure captures all possible edge interactions, and $\mathcal{G}_{1:K}$ is identifiable given identifiability of $\mathcal{G}^{1:T}$. $\square$

Notice that the assumption on each variable visiting all possible states is required if one aims to obtain the full structure of the conditional summary graph. Otherwise, if the state $k$ of a variable $\mathbf{x}_i$ is not visited, this leaves the outgoing edges of the $i$-th node in $\mathcal{G}_k$ undetermined, since there is no information to extract from the full time graph.

## B    IDENTIFIABILITY EXTENSIONS FOR STATE-DEPENDENT TIMINO

The first-order Markov assumption makes our definitions and identifiability analysis simpler. However, we argue this can still be very useful because (i) it usually suffices to describe particle trajectories (position, velocity, acceleration, etc), (ii) it allows us to present the conditional summary graph as a compact representation of the causal structure, and (iii) we present a method (SDCI) which leverages this assumption for efficient causal discovery. One could easily extend our identifiability results and definitions to higher order time-lagged effects. Incorporating contemporaneous effects is also possible, but requires further assumptions to consider an *identifiable functional model class* (IFMOC) (Peters et al., 2011) (e.g. linear $f_j^{s^{t-1}}$ with non-Gaussian disturbances).

## C    IMPLEMENTATION DETAILS

All the experiments are implemented in Pytorch (Paszke et al., 2019) and carried out on NVIDIA RTX 2080Ti GPUs.

### C.1    PROBABILISTIC FRAMEWORK FOR SDCI

This section aims to clarify our implementation in terms of the probabilistic modelling perspective. Note that some text from the main paper is re-used in this section. The underlying graphical model for conditionally stationary time series differs among scenarios. Here we illustrate the graphical models for scenario class 1 and 2.

### C.1.1    SCENARIO CLASS 1

**Generative model.**    Since the states $\mathbf{S}$ are observed at all times from our assumptions, we represent the joint distribution given some deterministic observed states $\tilde{\mathbf{S}}$.

$$p(\mathbf{X}, \mathbf{W}|\mathbf{S} = \tilde{\mathbf{S}}) = p_\psi(\mathbf{X}|\mathbf{W}, \mathbf{S} = \tilde{\mathbf{S}})p(\mathbf{W}) \tag{15}$$

where $p(\mathbf{W}) = \prod_{k=1}^K \prod_{ij} p_k(w_{ijk})$ represents the prior over conditional summary graphs (including edge-types). We note that $\mathbf{W}$ represents the set of graphs denoting the state-dependent interaction types among elements, and thus $p(\mathbf{W})$ is a categorical distribution. $p(\mathbf{W})$ can be used to provide domain knowledge by e.g. encouraging sparsity. The generative process of a sequence $\mathbf{X}$, factorises as

$$p_\psi(\mathbf{X}|\mathbf{W}, \mathbf{S}) = \prod_{t=0}^{T-1} \prod_{j=1}^N p_\psi(\mathbf{x}_j^{t+1}|\mathbf{x}^t, \mathbf{s}^t, \mathbf{W}) = \prod_{t=0}^{T-1} \prod_{j=1}^N \mathcal{N}(\tilde{\mathbf{x}}_j^{t+1}, \sigma^2 \mathbb{I}). \tag{16}$$

given some input states $\mathbf{S}$ and conditional summary graph $\mathbf{W} \sim p(\mathbf{W})$. The mean $\tilde{\mathbf{x}}_j^{t+1}$ is defined as follows. Given sampled conditional sumary graph $\mathbf{W} = \{w_{ijk}\}$, the decoder first queries the edge-type for element $j$ at time $t+1$ as $z_{ij}^t = w_{ijk'}$ for $s_i^t = k'$ (also see Eq. 3 in main paper). Then the information along the predicted edge-type interactions is retrieved and aggregated as follows [2]

$$\mathbf{h}_{ij}^t = \sum_{e>0} \mathbf{1}_{(z_{ij}^t=e)} f_e(\mathbf{x}_i^t, \mathbf{x}_j^t), \quad \tilde{\mathbf{x}}_j^{t+1} = \mathbf{x}_j^t + f_p\Big(\sum_{i\neq j} \mathbf{h}_{ij}^t, \mathbf{x}_j^t\Big), \tag{17}$$

where $\{f_e\}_{e=1}^{n_e-1}$ is a set of parametrizable functions, one defined for each edge type excluding the no-edge interaction. $f_p$ is a neural network that aggregates the information from the previous time-step, and models the dynamics for each variable $\mathbf{x}_j^{t+1}$.

**Inference.**    Note that we wish to compute the posterior over the conditional summary graphs, which might be intractable. We can introduce a variational distribution $q_\phi(\mathbf{W}|\mathbf{X}, \mathbf{S})$ that approximates the posterior for tractable inference and learning. The corresponding evidence lower bound (ELBO) is

$$\log p(\mathbf{X}|\mathbf{S} = \tilde{\mathbf{S}}) \geq -KL\Big(q_\phi(\mathbf{W}|\mathbf{X}, \mathbf{S} = \tilde{\mathbf{S}})||p(\mathbf{W})\Big) + \mathbb{E}_{q_\phi(\mathbf{W}|\mathbf{X}, \mathbf{S}=\tilde{\mathbf{S}})}\Big[\log p_\psi(\mathbf{X}|\mathbf{W}, \mathbf{S} = \tilde{\mathbf{S}})\Big] \tag{18}$$

---

[2]In training, $\mathbf{1}_{(z_{ij}^t=e)}$ is replaced by the $e$-th dimension of $w_{ijk'}$ which is sampled from $q_\phi$ with Gumbel-softmax relaxation.

Similar to Löwe et al. (2020), we use a factorized $q$ distribution $q_\phi(\mathbf{W}|\mathbf{X}, \mathbf{S}) = \prod_{k=1}^{K} \prod_{ij} q_\phi(w_{ijk}|\mathbf{X}, \mathbf{S})$. The encoder receives both $\mathbf{X}$ and $\mathbf{S}$ as the input, and extracts an embedding that represents the causal interaction conditioning on the state for every possible edge $\mathbf{x}_i \to \mathbf{x}_j$. We compute approximate posterior $q_\phi(w_{ijk}|\mathbf{X}, \mathbf{S})$ as follows,

$$\phi_{ij} = f_\phi(\mathbf{X}, \mathbf{S})_{ij} \in \mathbb{R}^{K \times n_\epsilon}, \tag{19}$$

$$q_\phi(w_{ijk}|\mathbf{X}, \mathbf{S}) = \Theta(\phi_{ijk}/\tau), \tag{20}$$

where $f_\phi(\mathbf{X}, \mathbf{S})$ is a neural network that returns outputs $\{\phi_{ij} : 1 \le i, j \le N\}$ given the input as $\mathbf{X}$ concatenated with a one-hot representation of the state variable $\mathbf{S}$. $\Theta(\cdot)$ denotes a softmax activation with temperature $\tau$. Note here the softmax activation is taken over the $k$-th row vector $\phi_{ijk} \in \mathbb{R}^{n_\epsilon}$ in $\phi_{ij}$, which is the edge embedding for $\mathbf{x}_i \to \mathbf{x}_j$ at state $k$. As this returns a categorical distribution for $q_\phi(w_{ijk}|\mathbf{X}, \mathbf{S})$, we apply the Gumble-softmax trick (Maddison et al., 2017) to enable direct differentiation for back-propagation during training.

The construction of $f_\phi(\mathbf{X}, \mathbf{S})$ is based on graph neural networks and MLPs. See Appendix C.2 for more details.

### C.1.2   SCENARIO CLASS 2

**Generative model.**   Note that now the states are not observed, but could directly inferred from observed data (as illustrated in Figure 1b). The corresponding joint distribution is as follows.

$$p(\mathbf{X}, \mathbf{W}, \mathbf{S}) = p_\psi(\mathbf{X}, \mathbf{S}|\mathbf{W})p(\mathbf{W}) = p(\mathbf{W}) \prod_{t=1}^{T} p_\psi(\mathbf{x}^t|\mathbf{x}^{t-1}, \mathbf{s}^{t-1}, \mathbf{W})p(\mathbf{s}^t|\mathbf{x}^t) \tag{21}$$

where we represent the time series structure for clarity purposes. $p(\mathbf{W})$, and the generative process are defined similarly as above, except now we also define the generation for $\mathbf{S}$.

**Inference.**   Again, we can use a variational approximation over the latent variables for tractable inference. We make a factorised approximation $q_\phi(\mathbf{W}, \mathbf{S}|\mathbf{X}) = q_\phi(\mathbf{W}|\mathbf{X})q_\phi(\mathbf{S}|\mathbf{X})$, where $q_\phi(\mathbf{W}|\mathbf{X})$ is defined in a similar way as in the fully-observed case (see above) except for using network $f_\phi(\mathbf{X})$ for the softmax logits. For the states, we consider $q_\phi(\mathbf{S}|\mathbf{X}) = \prod_{t=1}^{T} \prod_{i=1}^{N} q_\phi(s_i^t|\mathbf{x}_i^t)$, with each $q_\phi(s_i^t|\mathbf{x}_i^t)$ defined as

$$q_\phi(s_i^t|\mathbf{x}_i^t) = \Theta(\hat{s}_i^t/\gamma), \quad \hat{s}_i^t = \hat{f}_s(\mathbf{x}_i^t), \tag{22}$$

with $\gamma < 1$ as a temperature factor and $\hat{f}_s$ as a neural network. The ELBO is

$$\log p(\mathbf{X}) = \log \int p(\mathbf{X}, \mathbf{W}, \mathbf{S})d\mathbf{W}d\mathbf{S} \ge \mathbb{E}_{q_\phi(\mathbf{W}, \mathbf{S}|\mathbf{X})} \left[ \log \frac{p(\mathbf{X}, \mathbf{S}|\mathbf{W})p(\mathbf{W})}{q_\phi(\mathbf{W}|\mathbf{X})q_\phi(\mathbf{S}|\mathbf{X})} \right] \tag{23}$$

$$\ge -KL\left(q_\phi(\mathbf{W}|\mathbf{X})||p(\mathbf{W})\right) + \mathbb{E}_{q_\phi(\mathbf{W}, \mathbf{S}|\mathbf{X})} \left[ \sum_{t=1}^{T} \log \frac{p(\mathbf{x}^t|\mathbf{x}^{t-1}, \mathbf{s}^{t-1}, \mathbf{W})p_\psi(\mathbf{s}^t|\mathbf{x}^t)}{q_\phi(\mathbf{s}^t|\mathbf{x}^t)} \right] \tag{24}$$

$$\ge \sum_{t=1}^{T} \mathbb{E}_{q_\phi(\mathbf{W}, \mathbf{S}|\mathbf{X})} \left[ \log p_\psi(\mathbf{x}^t|\mathbf{x}^{t-1}, \mathbf{s}^{t-1}, \mathbf{W}) \right] - KL\left(q_\phi(\mathbf{W}|\mathbf{X})||p(\mathbf{W})\right) - KL\left(q_\phi(\mathbf{S}|\mathbf{X})||p(\mathbf{S}|\mathbf{X})\right) \tag{25}$$

In our implementation, we set $q_\phi(\mathbf{S}|\mathbf{X}) = p(\mathbf{S}|\mathbf{X})$, and thus $KL\left(q_\phi(\mathbf{S}|\mathbf{X})||p(\mathbf{S}|\mathbf{X})\right) = 0$. Note that we also sample states $\mathbf{S} \sim q_\phi(\mathbf{S}|\mathbf{X})$ using the categorical reparameterization (Maddison et al., 2017). Therefore, the decoding step can be performed as in Eq. 7 and the resulting message passing operation becomes

$$\mathbf{h}_{ij}^t = \sum_{k=1}^{K} \mathbf{1}_{(s_i^t=k)} \sum_{e>0} \mathbf{1}_{(w_{ijk}=e)} f_e(\mathbf{x}_i^t, \mathbf{x}_j^t), \tag{26}$$

and we apply similar Gumbel-softmax tricks as in the fully-observed state case to replace $\mathbf{1}_{(s_i^t=k)}$ and $\mathbf{1}_{(w_{ijk}=e)}$ during training.

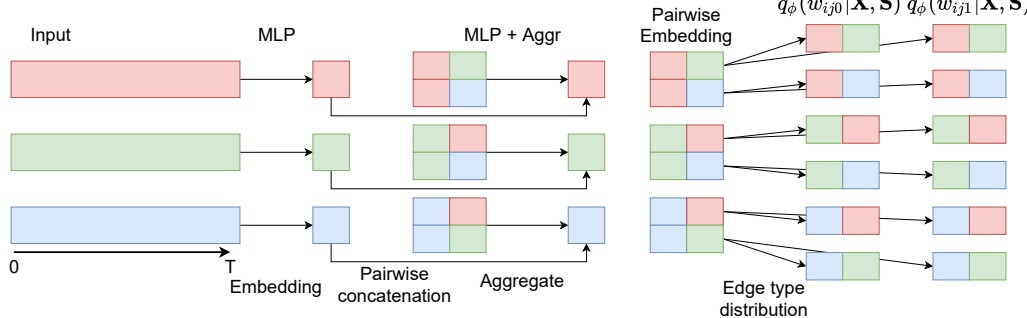

Figure 9: Illustration of the implementation of the SDCI encoder which is adapted from ACD (Löwe et al., 2020) and allow for conditioning on states. In the example, we consider 2 states.

## C.2 ENCODER ARCHITECTURE

Below we provide details of the encoder architectures.

**SDCI encoder** The first design of the architecture extends directly from ACD (Löwe et al., 2020) and we refer to this model as SDCI. We discuss the network construction for hidden state case as an example, for which the logits $\phi_{ij}$ for the distribution $q_\phi(\mathbf{W}|\mathbf{X})$ are obtained as follows. First, the model computes a latent embedding for each node $i$ using the whole sequence:

$$\mathbf{h}_i^1 = f_{\phi_1}(\mathbf{x}_i^{1:T}). \tag{27}$$

Then each embedding is updated using a graph neural network (GNN) that captures the correlations between nodes. Specifically the message passing procedure follows the two equations below:

$$\mathbf{h}_{ij}^1 = f_{\phi_2}(\mathbf{h}_i^1, \mathbf{h}_j^1), \tag{28}$$

$$\mathbf{h}_i^2 = f_{\phi_3}\left(\sum_{i \neq j} \mathbf{h}_{ij}^1\right). \tag{29}$$

Finally, we obtain the softmax logit $\phi_{ij} \in \mathbb{R}^{K \times n_\epsilon}$ for every possible edge $\mathbf{x}_i \to \mathbf{x}_j$ and every possible state $1 \leq k \leq K$:

$$\phi_{ij} = f_{\phi_4}(\mathbf{h}_i^2, \mathbf{h}_j^2). \tag{30}$$

The above network architecture design is visualised in Figure 9. according to equation 4. The details of the architecture settings follows the design in Löwe et al. (2020). Each embedding step $f_{\phi_i}$ uses two-layers of 256 dimensions and ELU (Clevert et al., 2016) activations followed by a batch normalization. $f_{\phi_4}$ uses skip connections and we modify its output size to generate a pairwise embedding for each of the $K$ states. For fully-observed state case, the architecture for $q_\phi(\mathbf{W}|\mathbf{X}, \mathbf{S})$ follows a similar structure, except that for the first layer we use $\mathbf{h}_i^1 = f_{\phi_1}(\text{concat}(\mathbf{x}_i^{1:T}, \mathbf{s}_i^{1:T}))$, where $\mathbf{s}_i^{1:T}$ is a set of one-hot vectors representing the states $\{s_i^t\}_{t=1}^T$.

## C.3 TRAINING SPECIFICATIONS (SDCI AND ACD)

All SDCI and ACD models have been trained using the following training scheme, including ACD (Löwe et al., 2020).

**Customized decoder (linear data only)** One of our objectives in the linear data experiments is to recover the underlying world parameters $\{\beta_k\}_{k=1}^{n_\epsilon-1}$. Therefore the decoder design imitates the message passing operation presented in Eq. 31, which allows us to initialize the decoder using the underlying world parameters and analyse the performance of the encoder as a separate entity from the whole model.

**Hyper-parameters** Following Kipf et al. (2018), the models are trained using ADAM optimizer (Kingma & Ba, 2015). The learning rate of the encoder is $5 \cdot 10^{-4}$, the learning rate of the decoder is $1 \cdot 10^{-3}$ for the synthetic linear data experiments and $5 \cdot 10^{-4}$ for spring data experiments. Learning rate decay is in use with factor of 0.5 every 200 epochs. We train for 1000 epochs in the linear experiments and 500 epochs in the springs experiments, using a batch size of 128. The decoder is trained with teacher forcing every 10 time-steps, i.e., it receives the ground-truth as input every 10 time-steps. The temperature $\tau$ is set to 0.5 and the variance of the Gaussian distribution of the decoder is $\sigma^2 = 5 \cdot 10^{-5}$. When considering the setting where we make the state dependent on the dynamics of the objects (scenarios 2 and 3), we set $\lambda = 10^3$. For inferring the hidden states, we set the temperature $\gamma = 0.1$ in the linear experiments and $\gamma = 0.05$ in the spring data experiments.

### C.4 Training specifications (VRNN)

The experiments with NBA player trajectories consider VRNN as a non-causal baseline to compare forecasting performance. Below we specify the network architecture and training scheme. To allow a fair comparison between SDCI, ACD, and VRNN, we modify the decoder defined in Chung et al. (2015) to condition the player positions on the ball features, similarly as we did for the previous models: $p_\theta(\mathbf{x}^t | \mathbf{x}^{<t}, \mathbf{z}^{\leq t}, \mathbf{b}^t)$, where $\mathbf{b}^t$ represents the ball features at time $t$. The architecture of the model follows the original work: 3-layer LSTM networks with 256 dimensions and a latent space size of 128 dimensions. The encoder and decoder architectures use two-layer MLPs of 256 dimensions. The models are trained using ADAM (Kingma & Ba, 2015) for 350K iterations with a learning rate of $10^{-4}$ and batch size 32.

## D Datasets

In this section we provide detailed information about the datasets used in this work. For linear and spring data, we generate 50000 samples of each setting for training the models. Regarding testing, we compute all the metrics using 10000 samples, except when comparing with CD-NOD, SAEM, and TdCM, where we use 200 samples since they require retraining the whole model for each sample. Moreover, notice that for these baselines we use sequences of much longer length (e.g., $T = 100$ or $T = 1000$) as these methods rely on longer sequences to get accurate results.

### D.1 Linear data

The ground-truth structural equation for the data generation process at time $t$ is

$$\mathbf{x}_j^{t+1} = \alpha \mathbf{x}_j^t + \sum_{i \neq j}^{N} \beta_k \mathbf{x}_i^t + \boldsymbol{\epsilon}_j^t, \quad k = \left( \mathcal{E}_{s_i^t} \right)_{ij}, \tag{31}$$

where $\alpha \in \mathbb{R}$ controls the self-connection, and $\boldsymbol{\epsilon}_i^t$ denotes independent noise sampled at each time-step.

Below we provide details of the data generation process for the linear data. First, we set the edge-type interactions. In our experiments we set $\alpha = 0.9$, $\beta_1 = 0.5$, and $\beta_2 = -0.5$ and $\epsilon_i^t \sim \mathcal{U}(-0.2, 0.2)$. To generate each sample, we need to sample the initial values of the continuous variable for each element, $\mathbf{x}_i^0 \sim \mathcal{N}(0, 2)$, and the underlying causal structure dependent on the state, $\mathcal{G}_{1:K}$. At each time-step, it suffices to query the edge-type $k$ for each pair of variables and apply the corresponding causal effect $\beta_k$ following Equation 31. The edge-type is $k = \left( \mathcal{E}_{s_j^t} \right)_{ji}$, where $(\mathcal{E}_s)_{ji}$ denotes the causal effect from $j$ to $i$ at state $s$, which has been defined at the beginning of the sequence. For all our experiments with this dataset, we simulate $N = 3$ variables. When considering hidden states in scenario class 2, we set $s_i^t = \mathbf{1}_{(|\mathbf{x}_i^t| > 2)}$ (2 states).

There are potential concerns that the generated samples produced in the linear data may be unstable. Still we use this data for one of the evaluations with the following reasons. First, they define a simulated environment where one has the ground truth and can debug and control simulation errors with ease. Furthermore, for one-dimensional variables $\mathbf{x}_i \in \mathbb{R}$ (which is our case), this dataset reduces to a first order Vector Autoregressive (VAR) model (Sims, 1980), which is widely used

in works related to causal discovery for time series data (Gong et al., 2015). The evolution of a sequence in this case can be expressed as follows:

$$\mathbf{x}^t = \mathbf{A}\mathbf{x}^{t-1} + \mathbf{e}^t \tag{32}$$

where $\mathbf{A}$ is the causal transition matrix and $\mathbf{e}^t$ is an independent noise process.

Regarding stability, the samples in this dataset are described by a causal transition matrix $\mathbf{A}$ where the diagonal elements are $\alpha$ and the off-diagonal elements are $\beta_k$ where $k$ is the edge-type interaction. For a first-order VAR to be stable, the singular values of $\mathbf{A}$ need to be smaller than one. Taking into the account that each sample can obey a different underlying causal graph, one needs to check this condition for all the possible arrangements of the off-diagonal elements (since the diagonal elements are always $\alpha$). The number of matrices that one needs to check grows rapidly for increasing number of variables, which makes the verification of this condition computationally infeasible (recall that computing the eigenvalues of a matrix has cubic cost $O(N^3)$). In practice, we generate random samples and keep them if the magnitude of the last observation is comparable to the initial one. We also require that the states are visited in similar proportions to ease the causal discovery task.

## D.2 SPRING DATA

When considering springs with directed connections, we follow the generation procedure described Kipf et al. (2018) with a small modification where the spring interaction between a pair of particles can change over time (depending on the state).

In this dataset, $N$ particles are simulated inside a 2D box where they can collide elastically with its walls. Each pair of variables is connected by a spring with uniform probability. To allow for identification of causal connections (directed edges), the connection is made unidirectional. The springs interact via the Hooke's law and this setting yield the following equations:

$$\mathbf{f}_{ij} = -\delta_k(\mathbf{r}_i - \mathbf{r}_j), \quad \ddot{\mathbf{r}}_i = \sum_{j=1}^{N} \mathbf{f}_{ij}, \quad \mathbf{x}_i = \{\mathbf{r}_i, \dot{\mathbf{r}}_i\} \tag{33}$$

where $\mathbf{f}_{ij}$ is the unidirectional interaction from particle $j$ to particle $i$, $\delta_k$ denotes the edge-type for each pair of variables, and $\mathbf{r}_i$ and $\dot{\mathbf{r}}_i$ denote the 2D position and velocity of each particle. The continuous variable $\mathbf{x}_i$ is constructed by concatenating the position and the velocity measurements.

Notice that the above equation defines the evolution of the continuous variable for a single time-step. In our setting, we have that $k = \left(\mathcal{E}_{s_j^t}\right)_{ji}$. Thus, $\mathbf{f}_{ij}$ will change over time, contrary to Kipf et al. (2018). Since we consider two edge-types, we define $\delta_0 = 0$ and $\delta_1 = 0.1$. To generate samples, we first generate a random conditional summary graph $\mathcal{G}_{1:K}$ and the initial location and velocity. Then, trajectories are simulated by solving the previous differential equations using leapfrog integration. The step size used is 0.001 and the trajectories are obtained by sub-sampling each 100 steps. In our experiments, we set $T = 80$. When considering hidden states in scenario class 2, we set $s_i^t = \mathbf{1}_{(\mathbf{x}_i^t > 0)}$ (2 states).

## D.3 NBA DATA

The NBA dataset (Linou, 2016) consists of recordings from 632 NBA games played during Winter 2015-2016. Each game is composed by approximately 400 to 600 events, which represents sequences of plays. In each trajectory, we find information about the ball location and 5 players of the 2 different teams (10 in total). The coordinates of the ball and players are represented in 3D and the length of each sequence can vary from 100 to 600. In our experiments, we consider a sequence up to 200 timesteps ($T = 100$ for reconstruction and the rest 100 steps

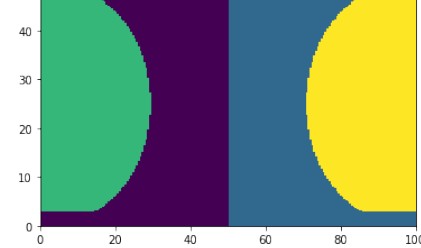

Figure 10: Hand-crafted ground-truth state map on NBA data which is forwarded to SDCI with observed states. Colours indicate different states.

---

[2]Data extracted from the following code repository `https://github.com/linouk23/NBA-Player-Movements` – last accessed 2022-09-28.

for prediction), which gives us a total training dataset of 150K samples and a test set of 6380 samples. Data inspection shows that the court size is $100 \times 50$, and we use this information to standardise the data. For experiments with SDCI with observed states, we design some ground-truth states which depend on different locations of the court. We set $K = 4$ and our choice is shown in figure 10.

## E  COMMENTS ON EVALUATION METRICS

### E.1  SUMMARY GRAPH IN TDCM

Huang et al. (2015) do not specify explicitly the computation of the summary graph (or an equivalent object). To allow a fair comparison, we take the same approach as in Huang et al. (2019), where and edge from $i$ to $j$ is not incorporated in the summary graph if the corresponding estimated time-varying coefficient has mean and variance lower than a theshold.

### E.2  ACCURACY OF THE SUMMARY GRAPH

To clarify, the evaluation of the summary graph considers correctly classifying the interaction between all pairs of nodes in both directions. To exemplify, for N=3 variables, our method needs to produce 6 predictions (one for every pair of edges in both directions). We then compute the accuracy of these predictions and average across all the samples in the test set.

### E.3  COMPUTING THE SUMMARY GRAPH IN SDCI

Notice that SDCI can extract the conditional summary graph (CSG) whereas the baselines we compare with only consider the summary graph (SG). Consequently, the only immediate way to compare the performance in capturing the causal structure among the methods we consider is to evaluate the latter. From the definition of summary graph, we deduce that one can estimate it by taking the union of the graphs in the CSG. This is used to compute the summary graphs of both SDCI and the ground truth structure of the generative process.

## F  ADDITIONAL RESULTS

In this section we report additional experiments and qualitative visualisations, which can be helpful to complement the main results from Section 4 in the main text.

### F.1  LINEAR DATA

**Scenario class 1**  We consider the case of scenario class 1 (see Section 1) where the states $\mathbf{S}$ are observed and their dynamics are independent from $\mathbf{X}$. We compare SCDI-Static with ACD, and evaluate the effect of explicitly modeling the underlying state. We further consider the case where the decoder is fixed and uses the ground-truth $\beta_k$ values, for which we denote as - FIXED DECODER or - FIXED DEC. for short.

Table 2: Test edge-type accuracy (in %) and MSE for linear data generated with scenario class 1 settings (fully-observed states) for 2 states and 2 edge-types.

| METHOD | EDGE ACCURACY | TEST MSE |
|---|---|---|
| ACD - FIXED DECODER | $66.02 \pm 0.29$ | $0.49 \pm 1.89 \cdot 10^{-2}$ |
| ACD | $66.44 \pm 0.29$ | $0.47 \pm 1.98 \cdot 10^{-2}$ |
| SDCI - FIXED DEC. | $90.43 \pm 0.23$ | $2.64 \cdot 10^{-2} \pm 4.55 \cdot 10^{-3}$ |
| SDCI | $93.84 \pm 0.19$ | $1.57 \cdot 10^{-2} \pm 4.03 \cdot 10^{-3}$ |

Table 2 shows the edge-type identification accuracy and test data reconstruction mean-squared error (MSE) for a simulation with $N = 3$ variables, $K = 2$ states, and $n_\epsilon = 2$ edge-types (no-edge and $\beta_1$). SDCI successfully performs the task of identifying the edges. In terms of the $\ell_1$ error for estimating $\{\beta_k\}$, ACD scores worse ($\sim 10^{-3}$) compared to SDCI ($\sim 10^{-5}$).

We repeat the same experiments with 3 edge-types and 2 states, and report the results in Table 3. As before, SCDI shows superior performance in comparison to ACD. In terms of the $\ell_1$ error for

Table 3: Test edge-type accuracy (in %) and MSE for linear data generated with scenario class 1 settings (fully-observed states) for 2 states and 3 edge-types.

| METHOD | EDGE ACCURACY | TEST MSE |
|---|---|---|
| ACD - FIXED DECODER | $49.29 \pm 0.31$ | $0.50 \pm 1.56 \cdot 10^{-2}$ |
| ACD | $34.11 \pm 0.28$ | $0.66 \pm 2.52 \cdot 10^{-2}$ |
| SDCI - FIXED DEC. | $92.75 \pm 0.22$ | $1.00 \cdot 10^{-2} \pm 1.57 \cdot 10^{-3}$ |
| SDCI | $84.36 \pm 0.28$ | $3.64 \cdot 10^{-2} \pm 2.19 \cdot 10^{-3}$ |

estimating $\{\beta_k\}$, we observe that SDCI provides more accurate estimations ($\sim 10^{-2}$ respectively) compared to ACD ($\sim 10^{-1}$).

**Scenario class 2** The results considering linear data with hidden states in scenario class 2 show that the CSG accuracy decreases when considering an additional edge-type (see Table 1). However, when leaving the summary graph constant (3-EDGE CONST) we obtain better CSG estimations. Figure 11 shows the approximate posterior $q_\phi(s_i^t|\mathbf{x}_i^t)$ inferred by SDCI in each of the 3 settings studied in this experiment along with the true underlying state function, $s_i^t = \mathbf{1}_{(|\mathbf{x}_i^t|>2)}$. We notice that SDCI achieves better results when the quality of the estimated state function is higher. This is expected, since querying the correct edge-type from the conditional summary graph strongly relies on accurate state estimations. Therefore, the model has to learn the state distribution and only then will be able to infer accurate causal structures. Although they can be challenging in general, our results show that SDCI is success-

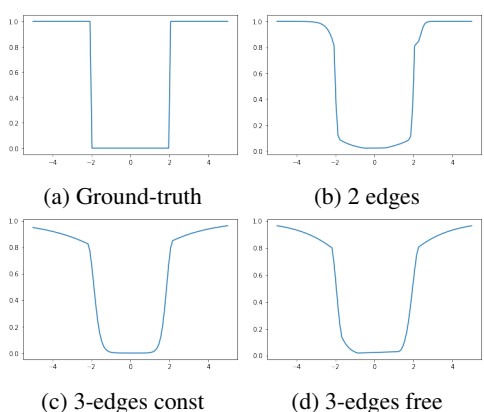

(a) Ground-truth      (b) 2 edges

(c) 3-edges const      (d) 3-edges free

Figure 11: Underlying state function for scenario class 2 in the linear dataset, $s_i^t = \mathbf{1}$.

ful in performing both learning steps in simple settings. This offers a promising direction of work towards the conditional stationary setting with hidden states.

Figure 12 shows samples for the linear data considering hidden states and 2 edge-types along with the corresponding causal summary graphs and summary graphs inferred by both SDCI and ACD. Since SDCI achieves high accuracy, the majority of the graph estimations match the true causal summary graph, which results in good forecasts. We also show two cases (two last rows) where the graph estimations do not match the ground-truth, and in these cases the model does not predict the future trajectories accurately. Similar to what we show in the main text, the trajectories obtained with ACD drop rapidly to 0, and the summary graph estimates are considerably worse.

## F.2 SPRINGS DATA

**Scenario class 2** We provide additional visualisations for the case with hidden states in Figure 13, where we show the predictions of both SDCI and ACD as well as the corresponding conditional summary graphs and summary graphs extracted by both methods respectively. As in the linear case, SDCI produces accurate causal graph estimates. Regarding time series forecasting, our method is able make reasonable predictions. Notice that to train the models, we use teacher forcing every 10 time-steps, which means that the learned models are less suited for long-term dynamics modelling. However, one can expect to obtain more accurate predictions by progressively reducing the teacher forcing frequency during training. Considering ACD, despite being restricted by assuming stationary time series, it still infers graph structures that allow the model to produce decent forecasts.

**Scenario class 3** We consider the case where the particles are contained in a box, and the state of a particle transitions when it collides with the wall of the box (see Figure 1c). For simplicity, we only consider $K = 2$ states that transition alternatively on wall collision. Due to the complexity of the state transitions, our method cannot identify accurate causal structures for hidden states. Instead,

Table 4: Test Edge-type accuracy (in %) and test MSE using spring data with state transitions on wall collision.

| Method | Edge accuracy | Test MSE | State Acc |
|--------|--------------|----------|-----------|
| ACD | $68.63 \pm 0.14$ | $1.46 \cdot 10^{-3} \pm 1.35 \cdot 10^{-5}$ | $98.21 \pm 0.02$ |
| SDCI | $79.19 \pm 0.14$ | $1.39 \cdot 10^{-3} \pm 1.33 \cdot 10^{-5}$ | $98.53 \pm 0.02$ |

we use the state information to learn the transitions and leave the hidden setting for future work. We report SDCI's performance along with ACD in Table 4. Regarding edge accuracy SDCI performs significantly better, and ACD is limited by considering stationary dynamics only. However, both methods achieve comparable test MSE metrics, indicating that that ACD can still make decent predictions even when it fails in identifying the edge-type interactions.

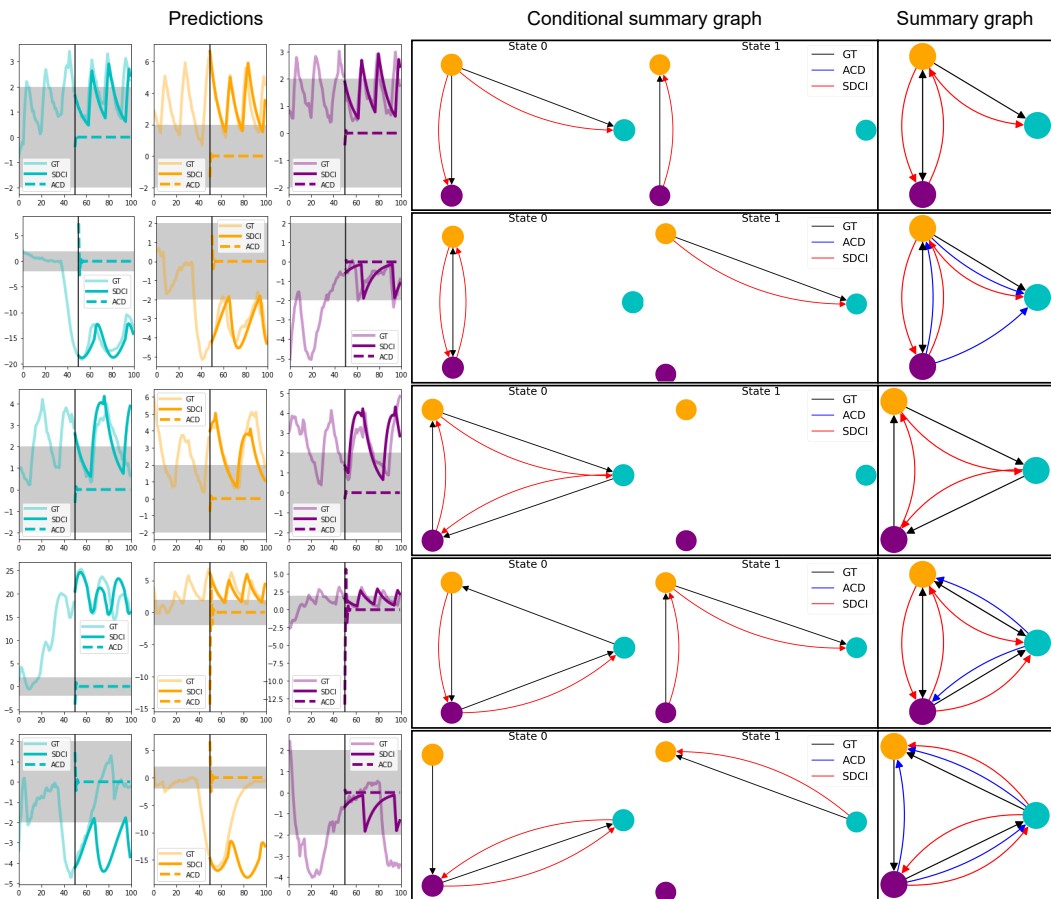

Figure 12: time series forecasting (left) of both SDCI (solid line) and ACD (dashed line) for 50 time-steps along with the ground-truth (GT, transparent line). The first 50 time-steps are given to the models as input and the background color represents the state value. We show the associated conditional summary graph (center) and summary graph (right) of SCDI (red) and ACD (blue) along with the ground-truth (GT, black) for each sample. Each row represents a different sample.

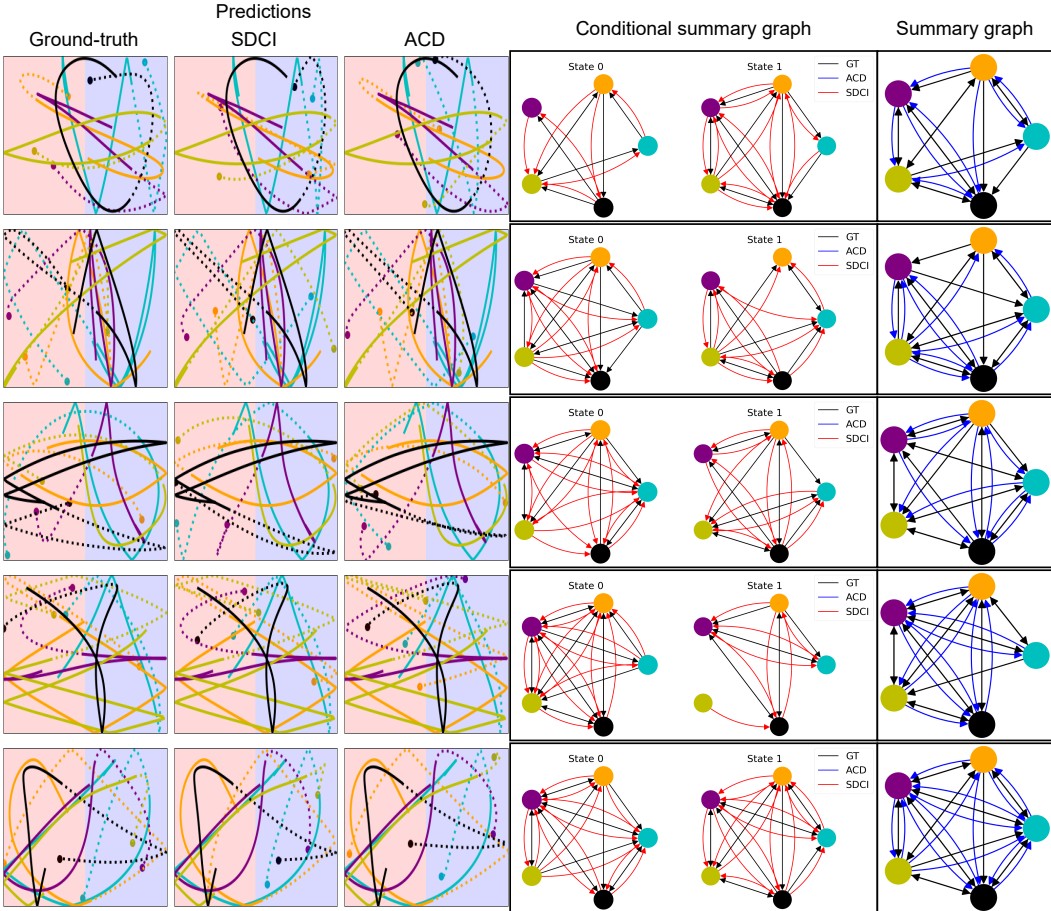

Figure 13: time series forecasting (left, dotted lines) of SDCI and ACD for 50 time-steps along with the ground-truth. We use solid lines to denote the input to the models and the background color represents the state value. We show the associated conditional summary graph (center) and summary graph (right) of SCDI (red) and ACD (blue) respectively along with the ground-truth (GT, black) for each sample. Each row represents a different sample.

