# OpenReview forum: "Causal discovery from conditionally stationary time series"
_ICLR.cc/2023/Conference — Submitted to ICLR 2023_

### Official Review · Reviewer_F9nk · 2022-10-24

**Confidence:** 4
**Correctness:** 4
**Technical Novelty And Significance:** 2
**Empirical Novelty And Significance:** 2
**Recommendation:** 5

**Clarity, Quality, Novelty And Reproducibility:**

Clarity: Very good. The paper is well presented.

Novelty: The novelty of this paper is not enough. For theoretical identifiability, it only considers the case when state variables are observed, which can be directly adapted from the previous literature.

Quality: Besides the above two points, the experimental verification fails to show the efficacy of the proposed practical estimation method.

**Strength And Weaknesses:**

Strength:

1. This paper considers an interesting and practically useful problem: time-delayed causal discovery from non-stationary data, and models the non-stationary behavior to be stationary conditioned on a set of state variables, which I think is a reasonable assumption.

2. The reported experimental results show that the proposed method has a better performance compared to other methods.

3. The organization and writing are good.

Weakness:

1. The authors only give the identifiability results when the state variables are observed (Scenario 1), which is trivial. Because of this, I fail to find significant contributions of this paper.

2. From Figure 4, it seems the practical performance of the proposed method decreases quickly with increasing variables and state values.

 Also, is the proposed method able to handle different time lags?



**Summary Of The Paper:**

This paper considers a particular setting for causal discovery from non-stationary time series. In particular, the non-stationary behavior is modeled as stationarity conditioned on a set of state variables.  The authors show the identifiability of their model, under the assumption that state variables are observed and there is no instantaneous causal effect. They further propose a VAE-based estimation method to estimate the causal graph, but without a consistency guarantee.

**Summary Of The Review:**

The paper is well presented, but the novelty is limited, especially in theoretical identifiability, and from the experimental results, the performance of the proposed method seems to decrease quickly with increasing variables and state values.

---

> ### Author Response · Authors · 2022-11-12
> **Response to reviewer (1/3)**
>
>
> We thank the reviewer for their constructive feedback on our work.
>
> Below we address the main concerns and **weaknesses**.
>
> > The authors only give the identifiability results when the state variables are observed (Scenario 1), which is trivial. Because of this, I fail to find significant contributions of this paper.
>
> We consider one should not appreciate the identifiability result as the only contribution of this paper. Still, we believe it is necessary to present some identifiability analysis, since it shows the implications of the assumptions our method relies on. Below we indicate some points on why our work is **significant**.
> - Our work considers causal discovery for non-stationary time series, where we find fewer established results as compared to the stationary case. Packed with our assumptions on conditional stationarity, we define the _conditional summary graph_ to efficiently represent the causal structure.
> - Similarly as indicated by the reviewer, we consider an “interesting and practically useful problem”, where we argue that the conditional stationary setting is worth studying, which based on our assumptions is identifiable.
> Note that if one just conditions the summary graph on the set of states, the result is a set of $K^N$ graphs due to the cartesian product of the states. We solve this problem by defining the _conditional summary graph_, which is a compact yet valid representation of the causal structure, which contains $K$ graphs. We consider that this representation is **significant** for its properties, e.g. being able to decompose a non-stationary time series into piecewise stationary ones.
> - We provide experiments on synthetic data, to empirically study the performance of our model on different environments and scenario classes. We also provide results in terms of data efficiency to show that our method does not require large-scale data to function. We also report generalization results to explore how our method handles unseen environments. We also show that our method **significantly outperforms** other baselines on causal discovery for time series.
> - We believe that there is a huge potential for realistic applications, which is showcased in our NBA experiment. We are able to show that SDCI discovers regimes which indicate the potential changes in dynamics in the court. For this reason, we consider that SDCI can be a suitable method for **data interpretability** in settings where one requires learning domain knowledge. The domain knowledge can be extracted by inspecting the learned regimes (as showcased in our paper), or the graph structures in the conditional summary graph (e.g. sparsity levels). We consider this point to be very strong in terms of significance, as there are few contributions which aim to discover causal structures on realistic domains (due to their non-stationary behavior).

---

> ### Author Response · Authors · 2022-11-12
> **Response to reviewer (2/3)**
>
> We continue addressing the **weaknesses**:
>
> >From Figure 4, it seems the practical performance of the proposed method decreases quickly with increasing variables and state values.
>
> We acknowledge the performance drops as one increases the number of state values and variables. We should note that in all the configurations, the number of observations and model capacity are held constant. From our results, we can extract several conclusions:
>
> - When considering a few variables (N=5), our method is able to maintain ~75% accuracy in edge identification. We argue this is still a valid result in comparison to ACD. Furthermore, we argue one reason for the performance drop is that as $K$ increases, the number of observations for each state value decreases (as we hold $T$ constant).
> - For increasing variables, our model needs to encode more information of the interactions for different states. A reasonable hypothesis is that the performance can be affected if the model capacity is constant as we increase the variables. However, we should note that for 10 variables and $K=2$, the performance is still reasonable.
> - We should not necessarily consider the performance drops on increasing variables and states as a bad result, since we need to put it in perspective with other baselines. This is why in our experiments we also report with ACD, which is consistently outperformed by our method.
>
> Furthermore, we would like to note that increasing variables is usually an unexplored setting in the majority of publications for causal discovery in time series. Below we provide a list with other contributions and their settings in terms of number of variables.
>
> - [Huang et al. 2019]: 5 variables.
> - [Huang et al. 2015]: 2 variables for synthetic experiments, 6 variables for real data.
> - [Huang et al. 2020]: 6 variables
> - [Lowe et al. 2022]: 5 variables (in a similar spring data)
> - [Hyvärinen et al. 2010] 5 variables
> - [Tank et al. 2018]: 20 variables with sparsity constraints (3 interactions for each variable at most). Note that we do not impose any sparsity constraint in our data, and thus increasing the complexity of the causal discovery task.
>
>
> In conclusion, we consider exploring performance on increasing variables an important issue for the potential applications . This way, one can learn the potential causes for a decrease in performance (e.g. low capacity, small number of time-steps $T$), and consider them when the ground-truth graph is unavailable in real domains.
>
>
> > Also, is the proposed method able to handle different time lags?
>
> We have not studied whether our method can handle this case since we formalize the problem in terms of a first-order Markov assumption. In Appendix B we motivate the use of the first-order Markov assumption. Below we summarize its motivation on why it can be useful:
> - It usually suffices to describe particle trajectories (position, velocity, acceleration, etc).
> - It allows us to present the conditional summary graph as a compact representation of the causal structure.
> - We present our method, which leverages this assumption for efficient causal discovery and we show it has potential for applications in real scenarios (see NBA experiments).
>
> It would be interesting to explore whether our method can handle different time lags by using GRUs (or similar) instead of MLPs in the decoder. However, for this case the method would not be in accordance with the main assumptions of our data generating process where the conditional summary graph is defined.
>
> > Quality: Besides the above two points, the experimental verification fails to show the efficacy of the proposed practical estimation method.
>
> We would like to engage in further discussion for this point. We consider our method to be efficient in terms of estimating the underlying graph structure when compared to other baselines.
>
> Moreover, we perform experiments on real data to showcase the potential for applications of our method in terms of causality-driven methods for forecasting and data interpretability. We show that our method outperforms a well-established sequential generative baseline (VRNN) and a causal discovery algorithm based on stationarity (ACD). We also perform inspections on the learned regimes to showcase interpretability.
>
>
> We hope that with all the points above, the reviewer can reconsider their assessment, since we consider this approach has potential for causal representation learning in real non-stationary domains.

---

> > ### Author Response · Authors · 2022-11-19
> > **Additional preliminary experiments with sparsity**
> >
> > Dear reviewer,
> >
> > We performed some preliminary experiments where we assume some sparsity on the causal structure with the aim of controlling the complexity of the causal discovery task on increasing variables. The idea is to replicate a similar setting as in Tank et al. (2018) mentioned above, which operates on 20 variables with sparsity. Below we report some results with $K=2$, $T=80$, and scenario class 1. We also compare to ACD as a reference.
> >
> > | Method       				| $N=5$ | $N=10$ | $N=20$ |
> > |:-------------------|:---------:|:------------------:|:-----------------:|
> > |SDCI (ours)	| 97.11|	89.83	   |	86.26	|
> > |ACD (Lowe et al. 2022)	| 72.05	|	66.64	   |	62.91	|
> >
> > Note that for $N=5$ sparsity is not applied. The above results are preliminary. We are currently pushing our models to increase performance and will consider to include a complete batch of these experiments (i.e. increasing both variables and states) in the Appendix if the reviewer considers this would improve the quality of our work.
> >
> > As a reference, Tank et al. (2018) reports 86.6 ± 0.2 accuracy for short sequences ($T=250$) and  98.4 ± 0.1 for larger ones ($T=1000$). The cited results correspond to non-linear stationary time series data with similar sparsity levels to ours. We observe that the length of the sequence has an important effect on improving performance.

---

> ### Author Response · Authors · 2022-11-12
> **References (3/3)**
>
> References:
>
> Huang, B., Zhang, K., Gong, M., & Glymour, C. (2019, May). Causal discovery and forecasting in nonstationary environments with state-space models. In International conference on machine learning (pp. 2901-2910). PMLR.
>
> Huang, B., Zhang, K., & Schölkopf, B. (2015, June). Identification of time-dependent causal model: A gaussian process treatment. In Twenty-Fourth international joint conference on artificial intelligence.
>
> Huang, B., Zhang, K., Zhang, J., Ramsey, J. D., Sanchez-Romero, R., Glymour, C., & Schölkopf, B. (2020). Causal Discovery from Heterogeneous/Nonstationary Data. J. Mach. Learn. Res., 21(89), 1-53.
>
> Löwe, S., Madras, D., Zemel, R., & Welling, M. (2022, June). Amortized causal discovery: Learning to infer causal graphs from time-series data. In Conference on Causal Learning and Reasoning (pp. 509-525). PMLR.
>
> Hyvärinen, A., Zhang, K., Shimizu, S., & Hoyer, P. O. (2010). Estimation of a structural vector autoregression model using non-gaussianity. Journal of Machine Learning Research, 11(5).

---

### Official Review · Reviewer_QqiG · 2022-10-24

**Confidence:** 5
**Correctness:** 3
**Technical Novelty And Significance:** 1
**Empirical Novelty And Significance:** 2
**Recommendation:** 3

**Clarity, Quality, Novelty And Reproducibility:**

The presentation is clear. Novelty is minor given the existing time-series discovery algorithms.

**Strength And Weaknesses:**

The authors assume access to the states at all times. Then one can simply run the existing stationary causal discovery algorithms conditioned on each realization of the state.

It turns out this is exactly what the authors propose as well.

Unfortunately, this simple extension might not be a sufficient contribution for an ICLR publication.

**Summary Of The Paper:**

The authors extend the existing work in time series causal discovery from stationary data to conditionally stationary time series data given the state.

**Summary Of The Review:**

Comments on experiments:
"for a fair comparison we evaluate the identification accuracy of the summary
graph such that it only considers the exis tence of an edge rather than capturing the edge type."
Does this mean you are simply checking the graph skeleton? Could you explain why this is more fair?

In the other experiments also, it's not clear if the proposed method shows promise compared to the baseline.

In Section 4.3, could you explain why causal discovery is important for forecasting, which is a predictive task?

---

> ### Author Response · Authors · 2022-11-12
> **Response to reviewer (1/3)**
>
> We appreciate your time in reviewing our paper, and we hope we can engage in further discussion regarding our work.
>
> First of all, we would like to clarify some of the **contributions** of this work for which we believe is novel and significant.
>
> - **Causality for non-stationary time series**: Contrary to stationary time series data, there are fewer established results towards modeling non-stationarity. Furthermore, we find even less contributions for works regarding non-stationary underlying graph structures, as the majority of them consider constant causal structures with time-dependent effects or noise distributions. Our work proposes this novel direction using conditional stationarity.
> - **Conditional summary graph**: We believe the concept of conditional summary graph is a significant contribution -- a compact and efficient summary of non-stationary time series. Importantly, although each of the nodes has its own associated state taking values in {1, …, K}, the number of conditional summary graphs does not scale exponentially as the number of nodes (N) grows – in fact it only scales as $O(K)$.
> - We propose a probabilistic model, which efficiently amortises learning of the conditional summary graph. We later clarify that extending a traditional causal discovery method is infeasible due to exponential scaling of the summary graphs, as also discussed in our paper. Furthermore, our approach is still able to operate with unobserved states and carry out the causal discovery computations in general non-stationary time series. The latter is useful to showcase the applications of our proposed framework.
> - We believe that there is a huge potential for realistic applications, which is showcased in our NBA experiment. We are able to show that SDCI discovers regimes which indicate the potential changes in dynamics in the court. For this reason, we consider that SDCI can be a suitable method for **data interpretability** in settings where one requires learning domain knowledge. The domain knowledge can be extracted by inspecting the learned regimes (as showcased in our paper), or the graph structures in the conditional summary graph (e.g. sparsity levels). We consider this point to be very strong in terms of significance, as there are few contributions which aim to discover causal structures on realistic domains (due to their non-stationary behavior). As discussed in the Conclusions Section, paragraph 2, our method could be applied to study the causal structure in videos, where we could use auxiliary information (e.g. poses) as states to account for non-stationarity.
>
> Next, we would like to clarify some of the **weaknesses** outlined by the reviewer:
>
> > The authors assume access to the states at all times.
>
> We would like to clarify that while our identifiability result requires observing states at all times, we also study settings where they are hidden from data and require to be inferred. The latter is interesting from an application perspective, since one might not have access to auxiliary states in real data.
>
> > Then one can simply run the existing stationary causal discovery algorithms conditioned on each realization of the state.
>
> We disagree with this statement, since this is not what we are proposing in our framework. Our problem does not consider a single state, but a set of them (one for each variable, N in total). Then, if one simply extends the existing stationary causal discovery approaches, the result is a set of $K^N$ graphs due to the cartesian product of the states. We solve this problem by defining the _conditional summary graph_, which is a compact yet valid representation of the causal structure, which contains $K$ graphs. We consider that this representation is **significant** for its properties, e.g. being able to decompose a non-stationary time series into piecewise stationary ones.
>
> Note that in Section 3.1, paragraph 11 (right after **identifiability of state-dependent TiMINo**), we already clarify that a direct extension, while possible, is not practical as it requires fitting an exponential increasing number of models $K^N$. Therefore, we consider that such an extension is not simple. Finally we use the previous point to justify amortizing the causal discovery task by using VAE estimation. Extending a stationary causal discovery baseline to this setting is not a simple task either, as one may encounter several challenges in terms of implementation.
>
> We hope the reviewer can reconsider their assessment in terms of this statement.
>
> > Unfortunately, this simple extension might not be a sufficient contribution for an ICLR publication.
>
> We provided a list of items on why we consider our work is novel and significant, and we hope the reviewer can reconsider this work as a valid contribution for this venue.

---

> ### Author Response · Authors · 2022-11-12
> **Response to reviewer (2/3)**
>
> Below we address the **comments**.
>
> > Comments on experiments: "for a fair comparison we evaluate the identification accuracy of the summary graph such that it only considers the existence of an edge rather than capturing the edge type." Does this mean you are simply checking the graph skeleton? Could you explain why this is more fair?
>
> As mentioned before, the majority of baselined methods for non-stationary causal discovery consider edges to be invariant in time. Therefore, for a fair comparison, we use the summary graph (SG) accuracy as the evaluation metric. Note that for SDCI, we can compute the SG by taking the union of the conditional summary graph (this is commented in Appendix E.3).
>
> Considering this, it does not mean we are checking the graph skeleton, but evaluating the previous metric, which is an undirected graph. Note that the existence of an edge is a binary variable, whereas an edge-type can have $n_{\epsilon}$ classes as stated in the main paper. In our setting, an edge exists if the edge-type differs from 0.
>
>
> > In the other experiments also, it's not clear if the proposed method shows promise compared to the baseline.
>
> Our work considers several experiments where we perform comparisons to baseline methods.
>
> **Linear data, scenario 2.** Our method outperforms the baselines for “$K=2$”, “$K=3$ const”, and “$K=3$ free”. For $K=3$ const, we find ACD performs comparably as mentioned, since the resulting causal structures are invariant in time (this is mentioned in the paper). For “$K=3$ free”, we observe that TCM performs comparably to SDCI. We believe this is caused by the instability of the resulting transition matrices (if we regard the linear data as a vector autoregressive process), which challenges the amortised inference performed in SDCI.
> **Spring data, scenario 1.** Our method consistently outperforms ACD. We report ACD performance to compare our results to the stationary baseline.
> **Spring data, scenario 2.** Our method consistently outperforms ACD and CD-NOD.
> **NBA data.** Our method outperforms VRNN and the stationary baseline (ACD).
>
> Note that the general baseline comparisons can be performed in terms of scenario 2, where the state values need to be inferred from data. We would like to ask which are the points that are not clear from the baseline comparison so that we can implement the corresponding changes in revision.

---

> ### Author Response · Authors · 2022-11-12
> **Response to reply (3/3)**
>
> We continue addressing the **comments**:
>
> > In Section 4.3, could you explain why causal discovery is important for forecasting, which is a predictive task?
>
> We would like to address this point by raising two discussion points, which we find they might be valuable for the causality crowd.
>
> **Forecasting for causality**
>
> In our paper, we observe that forecasting is important for causal discovery, in the sense that one first estimates the underlying causal structure, and then computes the error based on predictions. Other methods formalize the problem in different ways, but in general they fit an autoregressive model which informs about some underlying causal structure. Considering this, it is unarguable that forecasting plays an important role in causal discovery.
>
> **Causality for forecasting**
>
> The question now is: Could causality do the same for forecasting?
>
> An important concern when implementing causal discovery in real domains is how one can assess the correctness of the graph structure, especially if one knows nothing/little about the domain of interest. One example is brain data, where if we implement a causal discovery method to study the brain connectivity regions, we have no way to distinguish how reliable the learned graph structures are. Again, this is where the forecasting task helps us, since accurate forecasts will indicate more reliable causal structures.
>
> In our work, we study a realistic scenario concerning NBA player movements, where we don’t have access to a ground-truth graph structure, but still want to exploit causality for **data interpretability**. In the experiments, we present two main findings:
> 1. Our method is able to surpass a well-established sequential generative baseline in forecasting.
> 2. By inspecting the learned regimes in the hidden state case, we observe that the changes in regions are consistent with typical strategies in NBA plays.
> As we mention in the main paper, our method based on causal discovery can be a competitive baseline for forecasting, and most importantly for learning domain knowledge which can be done by inspecting the learned regimes (as we show in the paper), or also the inferred graph structures.
>
> Another concern is generalization for out-of-distribution forecasts. To understand how causality can play an important role in generalisation, there is a crucial assumption one can pose. While the data distribution of the observations might differ across domains, it might be reasonable to assume that the possible interactions happening in both cases are shared. We consider this to favour causality-driven methods over general sequential baselines, as we will find similar graph distributions across domains (but different distributions in terms of observations). In our experiments (Figure 7c), we show that our method outperforms VRNN in prediction, while generalising on other unseen NBA matches.
>
> Finally, we provide an example study which would favour causality-driven methods when we require decisions based on forecasts. Consider the following scenario where:
> 1. We require trustworthy predictions (e.g. decision-making in finance).
> 2. We have some domain knowledge (e.g. we know how some features can interact)
> 3. Assume that the sample we have is out-of-distribution.
>
> In this case, any deep-learning-based forecasting baseline will provide some predictions, but since we are in out-of-distribution data, it is very likely that the predictions are wrong and therefore unlikely to be trusted. This is where we consider that causality-driven methods can be beneficial, since once inspect the estimated causal structures to see if they coincide with the domain knowledge that we have, thus providing trustworthiness to the predictions.

---

### Official Review · Reviewer_3oxu · 2022-10-25

**Confidence:** 3
**Correctness:** 2
**Technical Novelty And Significance:** 2
**Empirical Novelty And Significance:** 3
**Recommendation:** 5

**Clarity, Quality, Novelty And Reproducibility:**

There seem to be significant issues in clarity and novelty.

For the clarity front, the main probabilistic model that looks interesting is not clearly explained. In particular, I do not understand how GNN was applied. GNN is an approach to learning node embedding vectors, given a graph, which is typically assumed to be an undirected graph. In the present setting, however, the graph is unknown and directed. One possibility could be to iterate between graph estimation and GNN, but no clear-cut explanation is given.

Overall, it is hard to follow the proposed probabilistic causal learning model, and hence, it is hard to figure out what the true novelty is over the existing work. This is really unfortunate because if such an edge-independent Bernoulli-like model works for causal structure learning, it can be a major achievement.

For the novelty front, the assumption of being able to observe the state itself sounds like a very strong assumption. It is not readily clear to me how it is a significant innovation in light of existing works. Also, the novelty of the proposed probabilistic structure learning model is not very clear to me.


**Details Of Ethics Concerns:**

The paper itself looks quite solid, but a paper under the exactly the sample title seems to have been submitted, peer-reviewed, and published at a workshop.
<<REMOVED BY PC>>

Those two papers are not identical but very similar. That version is not cited in this ICLR version. I think this paper falls in the category of dual submission. Although the workshop website says it is" non-archival,"
<<REMOVED BY PCS>>

such a claim applies only when the proceedings are NOT formally published, which is unfortunately not the case. In fact, the official bibtex entry says "inproceedings" with a clear reference to the workshop and the paper pdf, meaning that the paper has already been indexed as a peer-reviewed paper.

<<REMOVED BY PCS>>





**Strength And Weaknesses:**

Strength
- Addresses a general problem of state-dependent causal graph learning.
- Proposed a generative model to probabilistically learn the causality graph.
- Conducted empirical evaluation to show a good performance.

Weakness
- The assumption of being able to observe the state variable sounds very strong. The relative novelty to the existing works might not be very significant.
- Presentation can be improved. The main probabilistic model is not fully explained. It is not clear how the authors extend the existing work.

**Summary Of The Paper:**

This paper proposes a state-dependent causal learning framework for multivariate time series. The goal seems to learn a state-dependent causal graph between multivariate time series. Since causal structures can change dynamically in practice, the targeted task is practically useful if the state is automatically inferred. In this paper, the authors assume that the states are fully observable. Under this setting, the task is to learn a causal graph for each of the K distinct states.

The authors introduce a VAE-based algorithm to learn the probability of getting an edge between a node pair. The model seems to follow an existing work, but the detail is not clearly given.


**Summary Of The Review:**

From what is written above, I cannot endorse the technical validity of the proposed framework. However, I encourage the authors to continue this line of research. If they significantly improve the presentation, in particular how the GNN part and the generative model are designed, it is going to be a good paper.

---

> ### Comment · Program_Chairs · 2022-11-05
> **-**
>
> The workshop is non-archival, and submitting to such does not violate ICLR policy.
> -PCs

---

> ### Author Response · Authors · 2022-11-12
> **Response to reviewer (1/2)**
>
> We appreciate your time in reviewing our paper and hope we can engage in further discussion.
>
> Below we address the **weaknesses** outlined by the reviewer:
>
> > The assumption of being able to observe the state variable sounds very strong. The relative novelty to the existing works might not be very significant.
>
> While we agree fully observable states is a strong assumption, we perform experiments on real NBA data using some “hand-crafted” states to showcase how this assumption can be used in real domains. We also explore the difference from assuming observed states in comparison to the hidden state setting.
>
> The assumption of fully observable states has other interesting applications. As discussed in our paper (Conslusions Section, paragraph 2), one could apply our method for causal analysis in videos by treating them as time series where the components are embeddings generated from the entities interacting in the scene. One could then use some auxiliary variable as the state value, e.g. pose. This example illustrates how SDCI could account for the non-stationarity of real video data.
>
> We should note that we also explore scenarios where the states are unobserved, since our formulation allows for this extension with some simple modifications. This setting is useful to compare SDCI with other causal discovery baseline methods. Furthermore, it offers a direct analysis of causality from non-stationary time series without the requirement of accessing an additional categorical variable of each element.
>
> > Presentation can be improved. The main probabilistic model is not fully explained. It is not clear how the authors extend the existing work.
>
> Our probabilistic model extends from related works that usually account for stationarity. Our contribution is to adapt previous formulation to amortise the inference of the _conditional summary graph_. We provide a description about the implementation of SDCI in terms of encoder, sampling, decoder and objective; which is the usual approach when describing a VAE approach. We illustrate our model with a figure to show how the different components are integrated for learning. We appreciate the reviewer’s concern in improving clarity and encourage them  to point out the exact parts that are unclear so that this can be further improved in revision.

---

> ### Author Response · Authors · 2022-11-12
> **Response to reviewer (2/2)**
>
> Below we address the reviewers concerns on **clarity** and **novelty**:
>
> >  the main probabilistic model that looks interesting is not clearly explained.
>
> We devote Section 3.2 Implementation in describing our probabilistic approach. We believe the approach is clear in terms of estimating the underlying causal structure, and then fitting it using the decoder dynamics. We provide information of the equivalent VAE objective for graphs, which shows how we optimise the parameters.
>
> _edit: see newer reply for clarification._
>
> > GNN is an approach to learning node embedding vectors, given a graph, which is typically assumed to be an undirected graph. In the present setting, however, the graph is unknown and directed.
>
> As mentioned in Section 3.2 we use GNNs to estimate the _conditional summary graph_ (including edge-types) following previous approaches (which assume stationarity). From this sentence and Eq 4, it should be clear how the GNN is being used in our implementation.
>
> For more details on how to generate embeddings with GNNs, we refer to Appendix C.1 in our paper. There we show how the GNNs are used in SDCI. We welcome the reviewer to check this section for further details.
>
> _edit: for the newer revision we refer to C.2._
>
> >  it is hard to follow the proposed probabilistic causal learning model, and hence, it is hard to figure out what the true novelty is over the existing work.
>
> We encourage the reviewer to state which parts of our presentation are unclear to improve this upon revision. In terms of novelty regarding implementation, our model offers a solution to amortise learning the _causal summary graph_, which results in a computational expensive problem if one considers extending traditional causal discovery approaches. We refer to  Section 3.1, paragraph 11 (right after **Identifiability of state-dependent TiMINo**) for more details. We continue discussing novelty in the next point.
>
> > For the novelty front, the assumption of being able to observe the state itself sounds like a very strong assumption. It is not readily clear to me how it is a significant innovation in light of existing works.
>
> We clarify some of the **contributions** of this work for which we believe is novel and significant.
>
> - **Causality for non-stationary time series**: Contrary to stationary time series data, there are fewer established results towards modeling non-stationarity. Furthermore, we find even less contributions for works regarding non-stationary graph structures, as the majority of them consider constant causal structures with time-dependent effects or noise distributions. Our work proposes this novel direction using conditional stationarity.
> - **Conditional summary graph**: We believe the concept of conditional summary graph is a significant contribution -- a compact and efficient summary of non-stationary time series. Importantly, although each of the nodes has its own associated state taking values in {1, …, K}, the number of conditional summary graphs does not scale exponentially as the number of nodes (N) grows – in fact it only scales as $O(K)$.  We should note the novelty is that we incorporate one state per variable (N in total), as opposed to one state for all the variables (1 in total).
> - We propose a probabilistic model, which efficiently amortises learning of the conditional summary graph and does not scale exponentially as mentioned before. Moreover, our approach is still able to operate with unobserved states and carry out the causal discovery computations in general non-stationary time series.
> - We provide experiments on synthetic data, to empirically study the performance of our model on different environments and scenario classes. We also provide results in terms of data efficiency to show that our method does not require large-scale data to function. We also report generalization results to explore how our method handles unseen environments. We also show that our method **significantly outperforms** other baselines on causal discovery for time series.
> - We believe that there is a huge potential for realistic applications, which is showcased in our NBA experiment. We are able to show that SDCI discovers regimes which indicate the potential changes in dynamics in the court. For this reason, we consider that SDCI can be a suitable method for **data interpretability** in settings where one requires learning domain knowledge. The domain knowledge can be extracted by inspecting the learned regimes (as showcased in our paper), or the graph structures in the conditional summary graph (e.g. sparsity levels). We consider this point to be very strong in terms of significance, as there are few contributions which aim to discover causal structures on realistic domains (due to their non-stationary behavior).
>
> Based on the points above, we hope the reviewer can reconsider their assessment on novelty, as we consider this work can have significant impact for causal representation learning in real domains.

---

> ### Author Response · Authors · 2022-11-18
> **Revision released with clarifications on probabilistic approach in Appendix**
>
> Dear reviewer,
>
> We have submitted a revision including some clarifications of the probabilistic model we proposed. You can find the alternate formulation in Appendix C.1. In the corresponding section, we first define the generative model, and provide the expression for the ELBO in terms of the variational posterior for tractable inference and learning.
>
> For context, we provide the reviewer's notes to justify the revision we submitted.
>
> > Overall, it is hard to follow the proposed probabilistic causal learning model, and hence, it is hard to figure out what the true novelty is over the existing work.
>
> > For the clarity front, the main probabilistic model that looks interesting is not clearly explained. In particular, I do not understand how GNN was applied.
>
> We believe that now, Appendices C.1 and C.2 clarify the proposed probabilistic approach and the GNN implementation respectively. We would appreciate if the reviewer can verify if our paper is clear in terms of the implementation.

---

### Official Review · Reviewer_Hjuv · 2022-10-31

**Confidence:** 3
**Correctness:** 3
**Technical Novelty And Significance:** 2
**Empirical Novelty And Significance:** 3
**Recommendation:** 6

**Clarity, Quality, Novelty And Reproducibility:**

quality: ok

clarity: many improvement needed

originality: the problem is original as far as I can tell.

**Strength And Weaknesses:**

Strength:
The empirical evaluation is convincing with strong performance.

Weakness:
Presentation could be improved, with many design choices left poorly explained.
Identification result is decoupled from the deep learning framework  with VAE, which makes the theoretical results not really useful.
The proposed method seem to be adopted from many existing related work. While being applied to a new problem, it is unclear what difficulties in just combing them.


Comments:
-Figure 1 Scenarios: I found these scenarios seems arbitrary set. For example, in S2, s_t only depends on x_t, but in S3, s_t depends on x_t and x_{t-1} (and s_{t-1}). What about any subset of the dependence in Senario 3? Similarly, in S1 and S3, s_t depends on s_{t-1}  but not in S2. These inconsistency should be explained, if there is a specific choice behind it.
- Stationarity: the description is not totally clear. If one treat the joint distribution as a mixture distribution (with the states being the mixture variables), would it be stationary as well? If not, do you only assume the non-stationarity comes from s?
- is there a reason why auto-regressive connections are not considered?
- identiability: it would be good to have a formal statement on this in the paper. In addition, "at in this hidden state regime the
previous theoretical guarantees on identifiability no longer hold due to violations of our assumption" I'm not sure if the identifiability holds in the proposed VAE structure where you find the lower bounds.

**Summary Of The Paper:**

This paper proposes to discovery causal graphs in term of conditional summary graphs with conditional stationary time series data. Authors first present the framework for learning conditional time series data and then presented a VAE framework to instantiate the framework. Empirical study shows the superior performance over existing baselines.

**Summary Of The Review:**

it is a decent work to address a new problem with decent empirical performance, but technically it is not new and theoretical results are not consistent with the algorithm.

---

> ### Author Response · Authors · 2022-11-12
> **Response to reviewer (1/2)**
>
> We thank the reviewer for their time and constructive feedback and we hope we can engage in further discussions.
>
> First, we address the  **weaknesses** outlined by the reviewer:
>
> > Presentation could be improved, with many design choices left poorly explained.
>
> We thank the reviewer for further supporting this point with a list of clarification comments. We would also like to ask the reviewer to suggest which design choices are poorly explained so that we can include justifications in the appendix or the main paper. We address the comments in the next reply and we plan to include clarifications during revision.
>
> >  Identification result is decoupled from the deep learning framework with VAE, which makes the theoretical results not really useful.
>
> For this point, we refer to Section 3.1, paragraph 11 (right after **Identifiability of state-dependent TiMINo**), which is a key part in our paper. Below we provide some extra clarifications.
>
> One might think that approaching this causal discovery task with a VAE-based method is a cumbersome continuation of the identifiability theorem. However, it is a practical solution which follows from observing that an extension from TiMINo requires fitting an exponential number of models ($K^N$).
>
> We then observe that in a practical real-world scenario, it is reasonable to assume the functional components to be shared not only across data, but also across variables. Therefore, we find it justifiable to amortize the estimation of the conditional summary graph using VAEs.
>
> Regarding data distribution, note that there are differences between assumptions on the identifiability result and the VAE estimation method. The former is more general (as it is more flexible in the functional form given elements and states), and the latter is more restricted but still practical from an application perspective. The identifiability result being more general is good since it renders the underlying graphs structures identifiable in the VAE assumptions.
>
> Finally, we clarify that proving the consistency of our method is left as future work, as at this stage one would need to first revisit this problem for the stationary case. We support this point in our paper by referring to Geffner et al. (2022), which proves consistency for non-temporal causal discovery.
>
> Geffner, T., Antoran, J., Foster, A., Gong, W., Ma, C., Kiciman, E., ... & Zhang, C. (2022). Deep End-to-end Causal Inference. arXiv preprint arXiv:2202.02195.
>
> > The proposed method seem to be adopted from many existing related work. While being applied to a new problem, it is unclear what difficulties in just combing them.
>
> We agree our method can be viewed as an extension of previous causal discovery methods, i.e. _Amortized Causal Discovery (ACD)_.
>
> In terms of implementation, we outline the main problems one needs to face when adapting ACD for our setup and we leave the difficulty assessment to the reviewer’s judgment.
> - **Encoder**:
>     - Our implementation requires estimating the conditional summary graph as opposed to the summary graph, which from the encoder perspective consists of including an extra layer of dimensionality (to account for the states).
> - **Decoder**:
>     - The main challenge could come from computing the time-dependent edge-types at each time efficiently (in batch mode) during training. Note that we have $N$ state variables, therefore it is crucial to design this part efficiently.
> - **Hidden states**:
>     - When considering hidden states, we noticed some difficulty comes from designing a network (and training scheme) which doesn’t become too overconfident (or underconfident) in estimating the unobserved states during training.
>     - The previous challenge in computing the time-dependent edge-types efficiently now adds in another layer of complexity, as we now require the states to enable backpropagation to the decoder.
> In summary, the main challenge comes from integrating the state-dependent mechanisms in the decoder dynamics, and later extending this to the unobserved case while enabling direct backpropagation.
>
> Aside from the implementation challenges, we would like to remark that our main concern is to study the problem of conditionally stationary time series and motivate the potential for applications that such assumptions can have in terms of **interpretability** and reasoning in data where one lacks **domain knowledge** in general non-stationary data. For this reason, we include an experiment with real NBA player movements. In this case, our method discovers regimes for potential changes in dynamics on a player if one considers typical strategies in NBA plays.

---

> ### Author Response · Authors · 2022-11-12
> **Response to reviewer (2/2)**
>
>
> Below we address the **comments**:
>
> > Figure 1 Scenarios: I found these scenarios seems arbitrary set. For example, in S2, s_t only depends on x_t, but in S3, s_t depends on x_t and x_{t-1} (and s_{t-1}). What about any subset of the dependence in Senario 3? Similarly, in S1 and S3, s_t depends on s_{t-1} but not in S2. These inconsistency should be explained, if there is a specific choice behind it.
>
> Scenarios 1, 2, 3, and 4 are examples to illustrate the levels of complexity one can have given the dependency of the states. In the main paper, we provide examples to illustrate each level of complexity. Contrary to the others, the first scenario considers observed states at all times. The main feature is that the states are independent of the observations, and since they are observed, they could be either dependent or independent among them. In the example (figure 1), we make it dependent since we experiment with the states transitioning into increasing values. However, we also consider observed states in the NBA experiments, and they are hand-crafted in a scenario-2-like fashion. Therefore, the state dependencies in scenario 1 are not important as we are not modeling their dynamics. We can include a clarification of the latter during revision.
>
> > Stationarity: the description is not totally clear. If one treat the joint distribution as a mixture distribution (with the states being the mixture variables), would it be stationary as well? If not, do you only assume the non-stationarity comes from s?
>
> We here refer to non-stationarity in terms of the causal structure. We use the concept of non-stationarity to make a distinction with what Peters et al. (2017) [1] regards as a stationary time series (Chapter 10), i.e. considering edges being invariant in time. Our approach contains a non-stationary full time graph, since the edges are no longer invariant in time.
>
> We acknowledge that the non-stationarity of the full-time graph does not imply non-stationarity of the observations. It is true that a stationary sequence of states could produce a stationary distribution, although this is not necessarily happening.  We should clarify in revision that our concern about non-stationarity is in terms of the full-time graph connectivity, and not the distribution.
>
> > is there a reason why auto-regressive connections are not considered?
>
> In our work, auto-regressive connections are considered in terms of a first-order Markov structure. We would appreciate it if you let us know which parts suggested this was not the case.
>
> > identiability: it would be good to have a formal statement on this in the paper.
>
> We assume a formal statement refers to a statement of **Theorem 1** accompanied by the assumptions on the structural causal model. We agree this could improve the presentation of the paper.
>
> > In addition, "at in this hidden state regime the previous theoretical guarantees on identifiability no longer hold due to violations of our assumption" I'm not sure if the identifiability holds in the proposed VAE structure where you find the lower bounds.
>
> As mentioned before, we do not prove that our VAE approach is consistent with the identifiability theory. The sentence here is to remind the reader that there are no general identifiability guarantees on the causal structure for the hidden state region. It does not refer to the method, but the underlying causal structure of the data.
>
> We should note that despite having no identifiability guarantees, the hidden state setting (scenarios 2 to 4) is worth exploring in the main paper in terms of applications to realistic data where one has no access to states, and comparing our method to other baselines.

---

### Author Response · Authors · 2022-11-18
**Summary**

We thank all the reviewers for their time and feedback on our work.

We would like to summarise this reviewing process by combining the reviewers’ assessments on our work in combination with some of our concerns.


**Novelty of the approach**
- Although reviewer **Hjuv** considers the approach is _“original”_, reviewers **QqiG** and **F9nk** consider the contribution is not sufficient: _”...novelty is minor…”_, _”novelty of this paper is not enough”_.
- We are concerned that the assessment on novelty from reviewers **QqiG** and **F9nk** is purely based on the identifiability result. While we agree that the result on itself is minor, it should be seen as part of solving the bigger problem we are presenting. Below is a list of items on arguments supporting the novelty of our approach.
    - Our work considers causal discovery for non-stationary time-series data, where there are far fewer established results as compared with the stationary case. In particular we assume the time-series is stationary conditioned on a set of discrete states, and introduce the concept of _conditional summary graphs_ to efficiently represent the causal structure of the conditionally stationary dynamics. We provide identifiability proofs as well as practical implementations with probabilistic models.
    - **Conditional summary graph:** We believe the concept of conditional summary graph is a novel and significant contribution -- a compact and efficient summary of non-stationary time series. Importantly, although each of the nodes has its own associated state taking values in {1, …, K}, the number of conditional summary graphs does not scale exponentially as the number of nodes (N) grows – in fact it only scales as $O(K)$.  As compared with other work, Yao et al (2022) only consider $K$ regimes of stationarity (i.e., all the nodes are in one of the K regimes at one time).
- Although we specify the differences between our approach and other existing work (e.g. regime-dependent causal discovery) in the _Related work_ section, we consider the novelty front could be further clarified by embedding this section with other recent works which use $K$ regimes of stationarity [Yao et al. (2022)], or others which consider piece-wise stationarity [Monti et al. (2020)]. We support this point by reading reviewer **3oxu**: “... not clear how the authors extend the existing work”.

**Significance**
- We strongly believe there is a great potential for applications regarding our framework for causal discovery, and thus making our contribution significant. In fact, we showcase with real data that our method is suited for discoverying graphs in non-stationary domains and modeling sequential data, achieving superior performance than standard sequential baselines and gaining **data interpretability**.
- Our considerations are acknowledged by some reviewers. **F9nk**: _“... an interesting and practically useful problem”_. **3oxu**: _“the targeted task is practically useful…”_.
- Regarding assumptions, reviewer **3oxu** considers _“the assumption of being able to observe the state variable sounds very strong”_. While we agree with the latter, we still believe the assumption is valid in terms of applications. In our paper we already showcase that domain knowledge can be leveraged to hand-craft some state values, which results in higher performance than the stationary baseline.



**Clarity**
- 2 reviewers describe our manuscript as _“clear”_ (**QqiG**) and _“well presented”_ (**F9nk**).
- Reviewer **Hjuv** considers _“presentation could be improved”_. We will address the corresponding comments from **Hjuv** to improve the presentation of our paper after clarifying which parts are unclear.
- Reviewer **3oxu** considers _“the main probabilistic model is not fully explained”_. We have included a new Appendix section (C.1) where we provide an alternate formulation of our probabilistic framework with the hope of clarifying this point.

**Experiments**
- Our method significantly outperforms strong baselines of causal discovery for time-series, including Amortised causal discovery (ACD) and CD-NOD.
- As mentioned before, we showcase the applicability of SDCI for discovering causal structures in real non-stationary data.
- We observe there are some opposing views with respect to the experiments in our work. Reviewer **Hjuv** considers _“The empirical evaluation is convincing with strong performance.”_. **3oxu**: _“Conducted empirical evaluation to show a good performance“_. However, **F9nk** considers _“the experimental verification fails to show the efficacy of the proposed practical estimation method”_.
- Furthermore, we are very concerned that none of the reviewers acknowledged the applicability of our method for discovering causal structures in real non-stationary domains. We consider this a very strong point in favour of our contribution, since this important issue is less explored in the causality literature.

---

> ### Author Response · Authors · 2022-11-18
> **References**
>
> Yao, W., Sun, Y., Ho, A., Sun, C., & Zhang, K. (2022). Learning Temporally Causal Latent Processes from General Temporal Data. International Conference on Learning Representations.
>
> Monti, Ricardo Pio, Kun Zhang, and Aapo Hyvärinen. "Causal discovery with general non-linear relationships using non-linear ica." Uncertainty in artificial intelligence. PMLR, 2020.

---

### Decision · Program_Chairs · 2023-01-20

**Decision:**

Reject

**Justification For Why Not Higher Score:**

Lack of adequate sophistication/novelty.

**Justification For Why Not Lower Score:**

N/A

**Metareview: Summary, Strengths And Weaknesses:**

The authors study an extension of the state of the art in causal discovery for time series from stationary to conditionally stationary time series data given the state. In this setting, as pointed out by the reviewers we could just run the existing stationary causal discovery algorithms conditioned on each realization of the state, which is precisely what the others suggest. I agree with the reviewers that there is not enough novelty here to merit an ICLR publication/presentation.